# Genetic architecture distinguishes tinnitus from hearing loss

Royce E. Clifford [1,2,11] ✉, Adam X. Maihofer[1,3,11], Chris Chatzinakos[4,5], Jonathan R. I. Coleman [6,7], Nikolaos P. Daskalakis [4,5], Marianna Gasperi[1,3], Kelleigh Hogan[1,3], Elizabeth A. Mikita [1,3], Murray B. Stein [3,8,9], Catherine Tcheandjieu[10], Francesca Telese [3], Yanning Zuo[3], Allen F. Ryan[1,2] & Caroline M. Nievergelt [1,3,11] ✉

Tinnitus is a heritable, highly prevalent auditory disorder treated by multiple medical specialties. Previous GWAS indicated high genetic correlations between tinnitus and hearing loss, with little indication of differentiating signals. We present a GWAS meta-analysis, triple previous sample sizes, and expand to non-European ancestries. GWAS in 596,905 Million Veteran Program subjects identified 39 tinnitus loci, and identified genes related to neuronal synapses and cochlear structural support. Applying state-of-the-art analytic tools, we confirm a large number of shared variants, but also a distinct genetic architecture of tinnitus, with higher polygenicity and large proportion of variants not shared with hearing difficulty. Tissue-expression analysis for tinnitus infers broad enrichment across most brain tissues, in contrast to hearing difficulty. Finally, tinnitus is not only correlated with hearing loss, but also with a spectrum of psychiatric disorders, providing potential new avenues for treatment. This study establishes tinnitus as a distinct disorder separate from hearing difficulties.

Tinnitus, defined as sound heard in the absence of an external source, is a heterogeneous disorder[1] affecting more than 740 million adults globally and perceived as a severe problem by more than 120 million people[2]. The medical specialty responsible for research and treatment of tinnitus has been debated for years[3], in large part because of its intimate association with hearing, cognitive, and psychiatric disorders. Patients seek otolaryngologists with a "ringing" in their ears[4]. Mental health workers treat tinnitus patients for its connection to depression, suicide ideation, sleep disorders, and anxiety[5–8], and even mild tinnitus is associated with an increase in attempted suicide[5]. Neurologists treat cognitive disorders, which are associated

with tinnitus[9,10], and point to changes in imaging in numerous brain regions, while electroconductive studies demonstrate an increased spontaneous firing rate in the auditory cortex associated with loss of inhibition in the auditory pathway[4,11–14]. Wave V of the auditory brain response (ABR) has demonstrated increased latency in constant tinnitus, indicative of delayed connectivity to the inferior colliculus within the auditory pathway[15,16].

Twin and adoption studies of tinnitus indicate a heritability of 31–43%[17–19], and the largest genome-wide association study (GWAS) to date on >150,000 participants from the United Kingdom Biobank (UKB) demonstrated a polygenic architecture with a SNP-based

[1]Veterans Affairs San Diego Healthcare System, Research Service, San Diego, CA, USA. [2]University of California San Diego, Division of Otolaryngology – Head and Neck Surgery, La Jolla, CA, USA. [3]University of California San Diego, Department of Psychiatry, La Jolla, CA, USA. [4]Harvard Medical School, Department of Psychiatry, Boston, MA, USA. [5]McLean Hospital, Center of Excellence in Depression and Anxiety Disorders, Belmont, MA, USA. [6]King's College London, NIHR Maudsley BRC, London, UK. [7]King's College London, Social, Genetic and Developmental Psychiatry Centre, Institute of Psychiatry, Psychology and Neuroscience, London, UK. [8]Veterans Affairs San Diego Healthcare System, Psychiatry Service, San Diego, CA, USA. [9]University of California San Diego, School of Public Health, La Jolla, CA, USA. [10]VA Palo Alto Health Care System, Palo Alto, CA, USA. [11]These authors contributed equally: Royce E. Clifford, Adam X. Maihofer, Caroline M. Nievergelt. ✉e-mail: r2clifford@health.ucsd.edu; cnievergelt@health.ucsd.edu

heritability ($h^2_{SNP}$) of 6% and identified 6 genome-wide significant (GWS) tinnitus loci[20]. Importantly, a high genetic correlation and inferred causation was found between hearing difficulty (HD), psychiatric traits such as major depressive disorder, and tinnitus. Other complementary GWASs of tinnitus in the same UKB population added 4 additional loci[21,22]. It is of note that genetic studies comparing tinnitus and hearing impairment agree on a high genetic correlation and until now, provide little evidence of distinguishing genomic factors[20,23].

For future precision clinical therapy, it will be important to separate the genomics of tinnitus from HD. Although mechanistic-driven subclassification of tinnitus and highly-correlated hyperacusis has been proposed, that information has not as yet been available in large GWAS studies and its value has not so far come to fruition[24,25]. The high epidemiologic association of the two disorders points to an initial cochlear source of injury, whether the etiology is aging, noise, trauma, or otherwise, followed by cochlear deafferentation[26], possibly through the loss of inner hair cell connections with spiral ganglion cells as an early event[27]. In the case of hearing loss, this deafferentation may drive neuroplasticity within the auditory pathway tonotopically to the auditory cortex, and is seen both on the audiogram as frequency specific and on diffusion tensor imaging (DTI) most commonly in the auditory cortex and inferior colliculus[28].

On the other hand, tinnitus may lead to different changes which will require coordination with genomic findings. Increased delta band activity is seen in the thalamus and auditory cortex[29]. The inferior colliculus shows changes ranging from hyperactivity secondary to decreased GABAergic inhibition, to altered levels of glutamic acid decarboxylase[30]. Imaging studies reveal changes in resting state and sound-evoked BOLD fMRI responses in hippocampus, insula, and prefrontal cortex, among others, indicative of changes in connectivity both within and outside the auditory pathway[31]. These neuroplastic changes occurring in the presence of tinnitus may indicate genetic variation at play in parts of the brain associated with perceptions, emotions, and cognition.

Here we have tripled the sample size of previous tinnitus GWAS and expanded analyses beyond UKB and those of European ancestry. Applying novel analytic tools, we present a detailed characterization of the genetic architecture of tinnitus, disentangling it from hearing loss, and addressing its connection with psychiatric disorders and health-related traits. Our findings support the cross-disciplinary nature of

tinnitus, placing the disorder within the purview of multiple medical specialties.

## Results

### Tinnitus GWAS, meta-analysis and functional integration in European ancestry

Tinnitus was defined based on a combination of self-report and ICD codes, and GWAS was performed in two large biobanks, the UKB (173,015 EA subjects) and MVP (423,910 subjects of EA, AA, and LAT race/ethnicity), separately for each cohort and ancestry group (Supplementary Data 1). Details on the cohorts, phenotypes, and analyses are provided in the Supplementary Methods. A meta-analysis across EA cohorts, including 481,874 individuals, identified 29 genome-wide significant (GWS) loci (Fig. 1a, Supplementary Data 2). Quantile-quantile plots indicated inflation of test statistics ($GC_\lambda = 1.365$), (Supplementary Fig. 1a), of which 94.3% was accounted for by polygenic effects (based on the LDSC intercept of 1.026; SE = 0.01).

Of note, locus 22 corresponds to the well-known 8p23.1 inversion region, a polymorphism including multiple disease-associated genes with mRNA levels associated with the inversion genotype[32]. The inversion polymorphism has been associated with psychiatric and behavioral traits such as neuroticism[33] and obesity-related diseases[34]. We therefore determined inversion carrier status and found that the association of the locus 22 lead SNP was attenuated after stratification by inversion status (rs13252982: $p = 0.0015$, Supplementary Data 3A). No variant in the risk locus remained GWS after conditioning on inversion status. However, there was a significant association between tinnitus and the inversion polymorphism itself ($z = 5.49$, $p = 4.11 \times 10^{-8}$, Supplementary Data 3B), both in the MVP and UKB cohorts.

Functional mapping and annotation of the risk loci using the FUMA pipeline, and fine-mapping based on Polyfun+SUSIE are summarized in Supplementary Data 4 and Supplementary Fig. 2 (Circos plots of chromatin interactions and eQTLs). Due to the elevated LD within the 8p23.1 inversion, which limits fine-mapping of the 31 genes predicted to be associated with this locus, locus 22 was excluded in downstream analyses. We identified 78 protein-coding genes and some evidence of functionality of the 2,883 SNPs in LD with the 28 risk loci based on CADD scores, RegulomeDB scores, chromatin state and interactions using Hi-C data in neuronal cell lines across all loci. Fine-mapping reduced the genomic area implicated by >40% and reduced

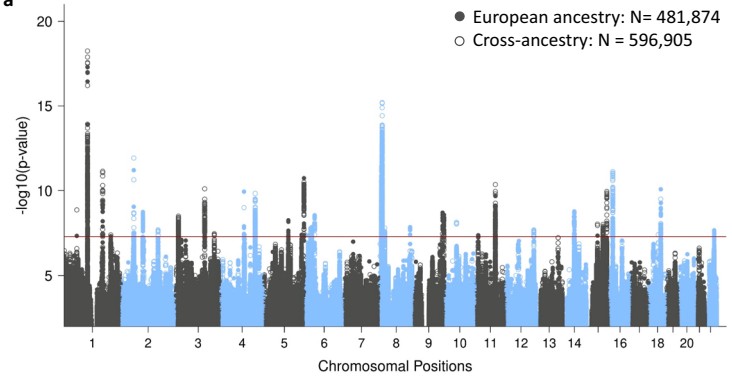

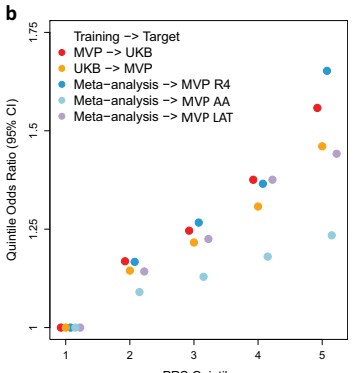

**Fig. 1 | Risk locus discovery and polygenic risk score analyses for tinnitus across different ancestries and data sources. a** Overlaying Manhattan plots from meta-analyses of tinnitus GWAS, showing 29 genome-wide significant (GWS) loci for the European ancestry (EA; full circles), and 30 GWS loci for the cross-ancestry (hollow circles) analyses. The *y* axis represents -log₁₀ p-values from two-sided z-test for meta-analyses effect estimates. The red line represents genome-wide significance at $p < 5 \times 10^{-8}$. **b** Genetic risk score (PRS) predictions for tinnitus comparing different training and target data. The *y* axis represents tinnitus odds ratios relative to the lowest quintile of PRS. Cross-dataset predictions from the EA MVP to UKB

(red circles) and UKB to EA MVP (orange circles) indicate similar prediction accuracies. Using the EA GWAS meta-analysis as training data (full circles in **a**) to predict tinnitus in a non-overlapping sample from MVP release 4 (MVP R4, blue circles) indicates that individuals in the highest quintile have 65% higher odds to develop tinnitus than individuals in the lowest quintile. Lower prediction accuracies are found using the EA meta-analysis to predict tinnitus in subjects of non-European ancestry (Hispanics (LAT) (purple circles) and African (AA) (light blue circles)). Sample sizes are shown in Supplementary Data 8.

the credible set (defined as in LD with lead SNP with $r^2 > 0.6$) to 649 SNPs.

## Cross-ancestry GWAS meta-analysis and functional integration

In preparation for a cross-ancestry meta-analysis, GWASs were performed separately in the smaller and admixed MVP LAT (11,819 cases, 20,869 controls) and AA datasets (18,216 cases and 64,127 controls). We identified one locus in LAT, but no locus was GWS in AA (Supplementary Data 5, Supplementary Figs. 1b & c and 3). The cross-ancestry meta-analysis identified 30 risk loci, including 162 protein-coding genes ($N$ = 596,905 EA, LAT and AA subjects, Fig. 1a, Supplementary Data 6–7). A comparison of risk loci across all 4 GWASs identified 39 non-overlapping tinnitus risk loci, including 9 additional loci identified in the cross-ancestry analysis (Table 1). Regional association plots comparing ancestry-specific analyses for the 39 loci are shown in Supplementary Data 21.

## PRS analysis

As a first step towards clinical applications of these tinnitus GWAS results we performed genetic risk score predictions (PRS) for tinnitus in independent non-overlapping EA target populations using PRS-CS[35]. Using MVP as training sample, the tinnitus PRS explained 1% of the phenotypic variation in the UKB target sample (liability scale, assuming a 12.5% population prevalence)[36], and similarly, the UKB-based PRS explained 0.7% of the variability in MVP (Supplementary Data 8). Using the combined EA GWAS meta-analysis as training set, 1.2% of the tinnitus variance was predicted in an independent MVP sample from release 4 (MVP R4), with individuals in the highest quintile having a 65% higher odds of tinnitus than individuals in the lowest quintile (Fig. 1b). These results show a significant improvement ($p < 1 \times 10^{-20}$) over the existing UKB-based PRS[20]. As observed for other traits[37], prediction accuracy of the EA-based tinnitus PRS is significantly lower in the LAT (0.6%) and AA (0.1%) cohorts.

## Gene-based analyses and identification of tinnitus-relevant tissues, pathways and drugs

As an alternative strategy to SNP-based analyses, gene-based analyses capture all of the potential risk-conferring variations within a gene and allow integration with gene expression data.

To aggregate genetic markers to the level of genes, we performed gene-based analyses using MAGMA. These analyses rely on matching reference data to accurately estimate SNP correlations (LD), which is complicated by local admixture[38]. We therefore focused our downstream analyses on EA only. Gene-based analyses identified 62 GWS genes, including 16 genes not identified by mapping approaches applied to the EA or cross-ancestry GWASs (Fig. 2a, Supplementary Data 9).

To identify the most likely tissues and pathways underling these gene-based associations, we performed a MAGMA tissue expression analysis including 30 general tissues, finding significant enrichment only in the brain. More specifically, in analyses including 54 tissues, 11 of 13 brain regions showed significant enrichment, most significantly in the cerebellar hemisphere (Fig. 2b, Supplementary Data 10). However, these analyses were based on GTEx v8 data, which do not include the human cochlea, a likely tissue of origin for tinnitus.

We therefore performed enrichment analyses based on available mouse data from two recent, large studies on cochlear cells[39] and the organ of Corti[40]. The cochlear data included 34 cell types found within circulating cells, glial cells, inner hair cells, lateral wall cells, neurons, supporting cells, and surrounding structures, and the organ of Corti included outer hair cells, inner hair cells, pillar, Deiter's, and melanocytes (see Fig. 2c for a diagram of the ear). In the MAGMA tissue expression analysis of cochlear cells, no cell type was significantly enriched when conditioned on the average expression of all cell types as the baseline, while the organ of Corti analysis showed a significant

enrichment ($p = 0.001$) of the outer hair cells (Fig. 2d, e and Supplementary Data 11). Using stratified LDSC as a complimentary enrichment analysis did not yield any significantly enriched cell types. These findings are consistent with the hypothesis that an outer hair cell injury is associated with tinnitus. While mouse expression highlights the role of the stria vascularis in hearing loss[41], tinnitus may have a more complex genetic basis both in the cochlea and the brain. Human cochlear tissue expression data, not currently available, will be necessary to confirm these findings, and to compare gene expression in the cochlea relative to other brain regions.

To see if specific biological pathways were implicated in tinnitus, a competitive set-based analysis of 15,485 pre-defined curated gene sets and GO terms obtained from MsigDB was performed. The curated_gene_sets: nikolsky_breast_cancer_14q22_amplicon was significantly expressed, a result driven mainly by 3 of the 14 genes, *GPR137C*, *TXNDC16*, and *NID2*, all in the credible gene sets of loci 31 and 32 (Supplementary Data 12).

Similarly, testing for enrichment of specific drug-classes and drug-sets was performed based on relevant drug-related databases[42–46]. Gene expression was significantly enriched for several drug classes, including muscle relaxants, antiepileptics, peripherally acting relaxants, anxiolytics, and benzodiazepine derivates. No specific drug sets were significant, but zolpidem and barbital, both acting on genes related to the GABA$_A$ receptor, included the leading associations (Supplementary Data 13-14). GABA and dopamine neurons work in concert, and activation of GABA neurons is associated with dopamine suppression. One of the significant genes, *GRK6*, regulates several types of dopamine receptors[47].

Consolidating results from these analyses, 23 genes were found with support from both the GWAS credible set and gene-based analyses (note: the chr.8 inversion, including 15 genes, is excluded here). The most significantly associated gene is *COL11A1*, a known non-syndromic and syndromic deafness gene that encodes type XIα collagen, identified within the lattice of the tectorial membrane, a gelatinous structure atop the Organ of Corti and in supporting Deiter's Cells (Fig. 2c)[48–50]. *NID2*, identified in previous hearing loss GWAS, binds to collagens I and IV in the basement membrane[41,51,52].

Six of the significant genes in this dataset are implicated in neural control within the context of growth and direction of neural processes, post-synaptic receptors, and post-synaptic membrane structures. *GRK6* ($p = 1.92 \times 10^{-11}$) is mediator of sensitivity to dopamine and has been identified in the lateral olivocochlear nucleus, part of an auditory reflex which dampens the cochlea's response to noise. *SHISA9* ($p = 1.40 \times 10^{-10}$) is an auxiliary subunit of AMPA-type glutamate receptors, a brain-specific transmembrane protein that engages in short-term plasticity. *PRR7* (gene-based meta-analysis $p = 5.75 \times 10^{-07}$) codes for a regulator of NMDA-mediated activity and is a synapse-to-nucleus messenger, promoting NMDA receptor-mediated excitotoxicity in neurons[4,5]. *GRM5* ($p = 6.45 \times 10^{-07}$) is a metabotropic receptor for glutamate, the chief excitatory neurotransmitter at the inner hair cell - spiral ganglion synapse. *DBN1* ($p = 2.35 \times 10^{-10}$) translates an actin-binding cytoskeleton-organizing protein, is engaged in dendrite neuroplasticity and is also found in pillar and Deiter's cells. *DNM1* ($p = 1.88 \times 10^{-08}$) is expressed in IHC and the post-synaptic membrane of spiral ganglion cells.

## Genetic architecture of tinnitus and comparison with hearing difficulty (HD)

We previously investigated genetic associations of tinnitus in the UKB and found a high genetic correlation with hearing difficulty ($r_g = 0.49$, CI: 0.35-0.63)[20]. Here we extended these analyses to include MVP and characterized the genetic architecture of tinnitus based on linkage disequilibrium score regression (LDSC) to estimate $h^2_{SNP}$ and $r_g$, and MiXeR, which uses univariate and bivariate Gaussian mixture modeling to estimate polygenicity and discoverability. We first compared the

**Table 1 | Comparison of 39 GWS tinnitus risk loci from GWAS in different ancestries**

| Locus | Lead SNP | Chr | Start[a] | Stop[a] | A1 | A2 | Cross-Ancestry (N = 596,905)[b] | | | EA (N = 481,874)[b] | | | AA (18,216 cases, 64,127 controls)[b] | | | | LAT (11,819 cases, 20,869 controls)[b] | | | |
|---|---|---|---|---|---|---|---|---|---|---|---|---|---|---|---|---|---|---|---|---|
| | | | | | | | A1 Freq | Z-score[c] | P value[c] | A1 Freq | Z-score[c] | P value[c] | A1 Freq | OR | SE | P value[c] | A1 Freq | OR | SE | P value[c] |
| 1 | rs35942154 | 1 | 56,097,025 | 56,171,731 | A | T | 0.084 | 6.064 | 1.33E-09 | 0.092 | 5.468 | 4.56E-08 | 0.033 | 1.043 | 0.035 | 2.29E-01 | 0.062 | 1.105 | 0.036 | 5.54E-03 |
| 2 | rs12722976 | 1 | 102,243,972 | 104,163,499 | G | C | 0.455 | 8.898 | 5.70E-19 | 0.491 | 8.651 | 5.12E-18 | 0.121 | 1.036 | 0.021 | 9.20E-02 | 0.506 | 1.031 | 0.018 | 8.07E-02 |
| 3 | rs4399218 | 1 | 170,170,094 | 170,196,129 | T | G | 0.337 | 6.853 | 7.26E-12 | 0.325 | 6.259 | 3.87E-10 | 0.430 | 1.018 | 0.012 | 1.50E-01 | 0.350 | 1.050 | 0.017 | 4.76E-03 |
| 4 | rs823154 | 1 | 205,651,560 | 205,799,987 | T | C | 0.366 | -5.490 | 4.01E-08 | 0.394 | -5.339 | 9.35E-08 | 0.199 | 0.994 | 0.015 | 6.83E-01 | 0.251 | 0.962 | 0.020 | 5.09E-02 |
| 4 | rs708727 | 1 | 205,651,560 | 205,799,987 | A | G | 0.369 | -5.294 | 1.19E-07 | 0.411 | -5.457 | 4.83E-08 | 0.076 | 1.016 | 0.024 | 5.03E-01 | 0.252 | 0.959 | 0.020 | 3.41E-02 |
| 5 | rs13016665 | 2 | 57,942,987 | 58,482,646 | A | C | 0.422 | -7.106 | 1.20E-12 | 0.426 | -6.874 | 6.25E-12 | 0.380 | 0.988 | 0.013 | 3.50E-01 | 0.446 | 0.964 | 0.017 | 3.39E-02 |
| 6 | rs3192177 | 2 | 98,325,330 | 98,623,406 | A | G | 0.320 | -5.237 | 1.63E-07 | 0.351 | -6.018 | 1.77E-09 | 0.120 | 1.016 | 0.019 | 4.01E-01 | 0.216 | 1.003 | 0.020 | 8.73E-01 |
| 7 | rs330634 | 2 | 150,949,434 | 150,974,693 | A | G | 0.141 | 1.284 | 1.99E-01 | 0.149 | 0.387 | 6.99E-01 | 0.070 | 0.965 | 0.028 | 2.05E-01 | 0.158 | 1.162 | 0.026 | 1.14E-08 |
| 8 | rs61541692 | 2 | 164,956,380 | 165,094,655 | C | G | 0.171 | 5.612 | 2.00E-08 | 0.164 | 5.281 | 1.29E-07 | 0.164 | 1.023 | 0.017 | 1.83E-01 | 0.289 | 1.029 | 0.019 | 1.26E-01 |
| 9 | rs307582 | 3 | 12,026,709 | 12,307,143 | T | C | 0.218 | -5.926 | 3.11E-09 | 0.214 | -5.329 | 9.85E-08 | 0.298 | 0.980 | 0.014 | 1.49E-01 | 0.140 | 0.936 | 0.027 | 1.48E-02 |
| 9 | rs900138 | 3 | 12,026,479 | 12,254,130 | A | G | 0.299 | -5.884 | 4.00E-09 | 0.262 | -5.608 | 2.05E-08 | 0.639 | 0.989 | 0.013 | 3.86E-01 | 0.230 | 0.958 | 0.022 | 4.98E-02 |
| 10 | rs16845806 | 3 | 127,710,474 | 128,064,118 | A | G | 0.134 | 6.502 | 7.94E-11 | 0.144 | 6.060 | 1.36E-09 | 0.056 | 1.041 | 0.032 | 2.09E-01 | 0.117 | 1.069 | 0.029 | 2.25E-02 |
| 11 | rs74505194 | 3 | 170,073,339 | 170,148,331 | G | A | 0.231 | 5.521 | 3.37E-08 | 0.199 | 4.956 | 7.18E-07 | 0.505 | 1.029 | 0.013 | 2.31E-02 | 0.231 | 1.020 | 0.020 | 3.28E-01 |
| 12 | rs13107325 | 4 | 103,001,649 | 103,198,082 | T | C | 0.071 | 6.114 | 9.73E-10 | 0.079 | 6.444 | 1.17E-10 | 0.014 | 0.962 | 0.053 | 4.57E-01 | 0.050 | 1.069 | 0.037 | 7.55E-02 |
| 13 | rs2214380 | 4 | 105,524,634 | 105,700,542 | G | T | 0.356 | -5.289 | 1.23E-07 | 0.334 | -5.492 | 3.98E-08 | 0.544 | 0.998 | 0.013 | 8.53E-01 | 0.342 | 0.985 | 0.019 | 4.10E-01 |
| 14 | rs6825241 | 4 | 152,223,904 | 152,752,413 | A | C | 0.458 | 6.409 | 1.47E-10 | 0.455 | 6.000 | 1.98E-09 | 0.522 | 1.013 | 0.012 | 2.87E-01 | 0.376 | 1.042 | 0.017 | 1.69E-02 |
| 15 | rs535664220 | 5 | 107,809,819 | 107,908,296 | INV[d] | T | 0.195 | -5.162 | 2.45E-07 | 0.164 | -5.831 | 5.50E-09 | 0.454 | 0.998 | 0.013 | 9.01E-01 | 0.193 | 1.022 | 0.022 | 3.05E-01 |
| 16 | rs57024367 | 5 | 166,155,566 | 166,175,717 | C | T | 0.175 | -4.916 | 8.82E-07 | 0.185 | -5.493 | 3.95E-08 | 0.063 | 0.985 | 0.028 | 5.84E-01 | 0.226 | 1.030 | 0.021 | 1.65E-01 |
| 17 | rs2545799 | 5 | 176,847,499 | 176,909,800 | G | T | 0.503 | 6.564 | 5.25E-11 | 0.541 | 6.716 | 1.87E-11 | 0.256 | 1.011 | 0.016 | 5.05E-01 | 0.368 | 1.016 | 0.020 | 4.35E-01 |
| 17 | rs2630763 | 5 | 176,847,499 | 176,909,800 | A | T | 0.495 | 6.633 | 3.30E-11 | 0.526 | 6.332 | 2.43E-10 | 0.317 | 1.029 | 0.014 | 5.04E-02 | 0.354 | 1.016 | 0.019 | 4.12E-01 |
| 18 | rs17769105 | 6 | 13,572,653 | 13,634,072 | A | G | 0.139 | -4.981 | 6.34E-07 | 0.153 | -5.480 | 4.27E-08 | 0.047 | 0.969 | 0.030 | 2.93E-01 | 0.090 | 1.052 | 0.030 | 9.44E-02 |
| 19 | rs9379893 | 6 | 26,467,182 | 26,929,518 | T | C | 0.202 | -5.649 | 1.61E-08 | 0.204 | -4.780 | 1.76E-06 | 0.215 | 0.948 | 0.015 | 2.49E-04 | 0.138 | 0.991 | 0.024 | 7.00E-01 |
| 20 | rs9257809 | 6 | 28,357,807 | 29,607,101 | G | A | 0.093 | -5.465 | 4.63E-08 | 0.105 | -5.015 | 5.30E-07 | 0.029 | 0.967 | 0.036 | 3.46E-01 | 0.027 | 0.888 | 0.051 | 2.04E-02 |
| 21 | rs1574430 | 6 | 43,260,011 | 43,397,259 | C | A | 0.541 | 5.633 | 1.77E-08 | 0.596 | 5.945 | 2.76E-09 | 0.116 | 1.016 | 0.020 | 4.44E-01 | 0.480 | 0.993 | 0.017 | 6.74E-01 |
| 21 | rs2242416 | 6 | 43,260,011 | 43,397,259 | A | G | 0.454 | -5.773 | 7.79E-09 | 0.400 | -5.884 | 4.02E-09 | 0.850 | 0.985 | 0.017 | 3.76E-01 | 0.562 | 0.996 | 0.017 | 7.96E-01 |
| 22 | rs34389419 | 8 | 8,088,230 | 11,830,150 | G | C | 0.420 | 8.083 | 6.34E-16 | 0.438 | 6.801 | 1.04E-11 | 0.334 | 1.058 | 0.014 | 1.06E-04 | 0.298 | 1.048 | 0.019 | 1.17E-02 |
| 22 | rs13252982 | 8 | 8,088,230 | 11,836,318 | G | C | 0.459 | 6.769 | 1.30E-11 | 0.496 | 7.564 | 3.90E-14 | 0.238 | 0.983 | 0.014 | 2.31E-01 | 0.293 | 1.016 | 0.019 | 3.84E-01 |
| 23 | rs7843128 | 8 | 22,447,426 | 22,542,962 | C | T | 0.368 | -5.655 | 1.56E-08 | 0.357 | -4.744 | 2.10E-06 | 0.489 | 0.972 | 0.012 | 2.17E-02 | 0.317 | 0.957 | 0.019 | 1.69E-02 |
| 24 | rs4999052 | 8 | 134,410,700 | 134,446,148 | C | T | 0.163 | 4.801 | 1.58E-06 | 0.128 | 5.675 | 1.39E-08 | 0.413 | 0.996 | 0.013 | 7.66E-01 | 0.241 | 0.974 | 0.020 | 1.79E-01 |
| 25 | rs3003615 | 9 | 130,982,416 | 131,015,279 | A | G | 0.493 | -5.517 | 3.45E-08 | 0.532 | -6.003 | 1.94E-08 | 0.121 | 0.992 | 0.020 | 6.74E-01 | 0.584 | 1.010 | 0.018 | 5.72E-01 |
| 26 | rs56303154 | 9 | 136,925,663 | 136,972,701 | T | C | 0.160 | 5.889 | 3.89E-09 | 0.174 | 5.799 | 6.69E-09 | 0.069 | 1.056 | 0.024 | 2.19E-02 | 0.113 | 0.981 | 0.026 | 4.55E-01 |
| 26 | rs1076411 | 9 | 136,925,663 | 136,974,020 | C | T | 0.217 | 5.227 | 1.72E-07 | 0.220 | 5.955 | 2.61E-09 | N/A[e] | N/A | N/A | N/A | 0.169 | 0.938 | 0.029 | 2.48E-02 |
| 27 | rs16915908 | 10 | 52,901,491 | 53,025,562 | C | T | 0.027 | -5.788 | 7.14E-09 | 0.020 | -5.204 | 1.96E-07 | 0.082 | 0.949 | 0.023 | 2.35E-02 | 0.031 | 0.938 | 0.052 | 2.19E-01 |
| 28 | rs12805064 | 11 | 13,230,633 | 13,266,951 | A | G | 0.486 | 5.319 | 1.04E-07 | 0.537 | 5.483 | 4.18E-08 | 0.131 | 1.020 | 0.019 | 3.01E-01 | 0.371 | 0.997 | 0.018 | 8.59E-01 |
| 29 | rs67279079 | 11 | 88,395,396 | 89,037,064 | T | C | 0.239 | 6.590 | 4.40E-11 | 0.268 | 6.359 | 2.03E-10 | 0.053 | 1.007 | 0.030 | 8.30E-01 | 0.134 | 1.077 | 0.026 | 3.73E-03 |
| 30 | rs883263 | 12 | 123,447,928 | 123,913,433 | A | G | 0.223 | -5.608 | 2.04E-08 | 0.217 | -4.846 | 1.26E-06 | 0.255 | 0.974 | 0.014 | 6.02E-02 | 0.256 | 0.953 | 0.020 | 1.45E-02 |
| 31 | rs27498882 | 14 | 52,502,345 | 52,516,601 | T | G | 0.426 | 5.906 | 3.50E-09 | 0.418 | 6.031 | 1.63E-08 | 0.419 | 1.007 | 0.012 | 5.81E-01 | 0.571 | 1.013 | 0.017 | 4.56E-01 |
| 32 | rs10151051 | 14 | 52,903,642 | 53,067,681 | G | A | 0.372 | 5.790 | 7.06E-09 | 0.340 | 5.909 | 3.43E-09 | 0.601 | 1.017 | 0.013 | 1.89E-01 | 0.443 | 0.996 | 0.017 | 7.99E-01 |
| 33 | rs1912629 | 15 | 47,995,384 | 48,108,119 | A | G | 0.074 | 5.746 | 9.14E-09 | 0.081 | 5.304 | 1.13E-07 | 0.018 | 1.062 | 0.052 | 2.49E-01 | 0.061 | 1.086 | 0.038 | 3.09E-02 |

**Table 1 (continued) | Comparison of 39 GWS tinnitus risk loci from GWAS in different ancestries**

| Locus | Lead SNP | Chr | Start[a] | Stop[a] | A1 | A2 | Cross-Ancestry (N = 596,905)[b] | | | EA (N = 481,874)[b] | | | AA (18,216 cases, 64,127 controls)[b] | | | | LAT (11,819 cases, 20,869 controls)[b] | | | |
|---|---|---|---|---|---|---|---|---|---|---|---|---|---|---|---|---|---|---|---|---|
| | | | | | | | A1 Freq | Z-score | P value[c] | A1 Freq | Z-score | P value[c] | A1 Freq | OR | SE | P value[c] | A1 Freq | OR | SE | P value[c] |
| 33 | rs17388803 | 15 | 48,027,204 | 48,108,119 | C | A | 0.106 | 5.712 | **1.11E-08** | 0.116 | 5.474 | **4.41E-08** | 0.029 | 1.044 | 0.040 | 2.81E-01 | 0.098 | 1.043 | 0.029 | 1.50E-01 |
| 34 | rs11631214 | 15 | 72,101,851 | 73,106,615 | T | A | 0.403 | -5.738 | **9.56E-09** | 0.366 | -5.091 | 3.57E-07 | 0.670 | 0.978 | 0.013 | 9.34E-02 | 0.477 | 0.963 | 0.017 | 2.68E-02 |
| 34 | rs7180191 | 15 | 72,103,427 | 72,656,241 | C | T | 0.371 | -5.598 | **2.17E-08** | 0.311 | -5.672 | **1.41E-08** | 0.847 | 0.993 | 0.019 | 7.30E-01 | 0.425 | 0.975 | 0.021 | 2.29E-01 |
| 35 | rs4453453 | 15 | 89,223,353 | 89,265,679 | G | T | 0.217 | -6.448 | **1.14E-10** | 0.222 | -5.372 | 7.78E-08 | 0.107 | 0.949 | 0.020 | 7.06E-03 | 0.342 | 0.950 | 0.018 | 4.34E-03 |
| 36 | rs4781383 | 16 | 13,021,889 | 13,118,299 | T | C | 0.258 | 6.845 | **7.62E-12** | 0.286 | 6.289 | **3.20E-10** | 0.072 | 1.042 | 0.024 | 9.30E-02 | 0.159 | 1.053 | 0.023 | 2.46E-02 |
| 37 | rs9967532 | 18 | 42,308,640 | 42,338,070 | T | G | 0.196 | 5.498 | **3.83E-08** | 0.188 | 4.890 | 1.01E-06 | 0.265 | 1.032 | 0.015 | 3.03E-02 | 0.187 | 1.032 | 0.024 | 1.85E-01 |
| 38 | rs613872 | 18 | 53,197,944 | 53,413,423 | G | T | 0.148 | -6.297 | **3.04E-10** | 0.166 | -6.493 | **8.43E-11** | 0.030 | 0.988 | 0.036 | 7.28E-01 | 0.091 | 0.972 | 0.029 | 3.20E-01 |
| 39 | rs35629137 | 22 | 40,540,762 | 40,720,963 | C | CA | 0.376 | -4.975 | 6.53E-07 | 0.410 | -5.600 | **2.15E-08** | 0.107 | 1.013 | 0.021 | 5.48E-01 | 0.348 | 0.999 | 0.018 | 9.72E-01 |

[a]Start and stop positions (in bp) for the ancestry highlighted.
[b]Results highlighted indicate leading SNPs for a specific locus and ancestry; bolded p values indicate p values below genome-wide significance threshold.
[c]Meta analysis and regression effect estimates were tested for significance using the two-sided z-test. Results bolded where genome-wide significant (p < 5 × 10⁻⁸).
[d]TTTTCCTAATGTCAG.
[e]Not imputed due to low imputation quality (Info score <0.6).

genetic architecture of the demographically different MVP and UKB cohorts and found strong similarities, with comparable heritabilities ($h^2_{SNP} = 0.063$ in MVP, SE = 0.005, liability scale to account for disease prevalence; 0.061 in UKB, SE = 0.004, observed scale for ordinal definition of tinnitus, respectively), and a high genetic correlation ($r_g = 0.834$, SE = 0.041; Supplementary Data 15A & B). In addition, while discoverability was slightly higher in the UKB than in MVP ($z = 2.52$, <0.012), there was no significant difference in MiXeR estimates for the number of causal variants needed to explain 90% of $h^2_{SNP}$ ($z = 0.24$, $p = 0.81$), and bivariate MiXeR analyses did not identify cohort-specific polygenicity (AIC = <0) (Fig. 3a). We thus performed downstream analyses on the combined tinnitus GWAS meta-analysis, which had a highly significant $h^2_{SNP} = 0.07$ (SE = 0.004, on the liability scale).

We next compared the genetic architecture of tinnitus with self-reported hearing difficulty (HD). Tinnitus was significantly more polygenic ($z = 11.6$, $p < 4.12 \times 10^{-31}$) than HD (Supplementary Data 15C), with 9,466 variants influencing tinnitus compared to 4,038 variants for HD. A bivariate MiXeR model was a better fit than a minimum model of genetic correlation (best vs. min AIC = 0.57, Supplementary Data 15D), indicating polygenic overlap beyond what would be expected based on genetic correlation. The $r_g$ estimated across all variants was 0.59 (SE = 0.01) and the genetic correlation among shared variants between tinnitus and HD was 0.94 (SE = 0.05). A large number of causal variants are shared between HD and tinnitus ($N = 3,850$), and shared variants accounted for the majority (95.4%) of the variants influencing HD, but only 40.7% of the variants influencing tinnitus (Fig. 3b). In regards to phenotype-specific variants for tinnitus and HD, the bivariate MiXeR model, which allows for phenotype-specific variation, was a better fit than a model that assumes that all of the variants of the less polygenic phenotype are a subset of the more polygenic phenotype (best vs. max AIC = 0.38). In particular, MiXeR estimated 5,615 tinnitus-specific variants, compared to 188 HD-specific variants. These findings were similar in both the MVP and UKB cohorts, with fewer causal variants and only a very small unique polygenic component for HD compared to tinnitus (Supplementary Data 15D).

### Identification of tinnitus-specific risk loci

While bivariate Gaussian mixture modeling estimates the overall number of unique and shared causal variants between traits, it does not identify specific risk loci for these traits. We therefore performed additional GWASs to identify risk loci specifically for tinnitus from shared risk loci for HD. Since information on hearing difficulty was available at the subject level, we first adjusted the tinnitus GWAS for HD. This covariate adjustment attenuated the effects of all 29 tinnitus risk loci (8 remain GWS) and identified 2 new GWS tinnitus loci on chromosome 7 and 12 (Supplementary Data 16, Fig. 3c & d). Several genes map to the locus on chromosome 12, including *CBX5* and *NFE2*, both identified predominantly in IHC and pillar cells[49]. As an alternative approach to identify tinnitus-specific risk loci, we also performed a case-case GWAS (CC-GWAS[53]), comparing tinnitus cases versus HD cases. CC-GWAS identified 10 GWS loci (Fig. 3e, Supplementary Data 16), of which 2 loci show a tinnitus-specific effect (mapping to locus 8 and 36). An additional locus was identified on chromosome 11 in *LUZP2* with opposite effect in tinnitus and HD, predominantly seen in pillar cells with little or no expression in the cochlea otherwise. *LUZP2* is expressed in the brain and spinal cord and is downregulated in gliomas[54]. Overall we identified more HD-specific risk loci than tinnitus-specific risk loci. This is consistent with the higher discoverability of HD compared to tinnitus (see Supplementary Data 15C). To further characterize these tinnitus and HD-related loci we also performed phenome-wide association studies (PheWAS) across an extensive set of phenotype domains for each leading SNP using the GWAS catalog implemented in FUMA (Supplementary Data 17).

A genome-wide comparison of the tissues underlying these associations based on MAGMA tissue expression analyses further

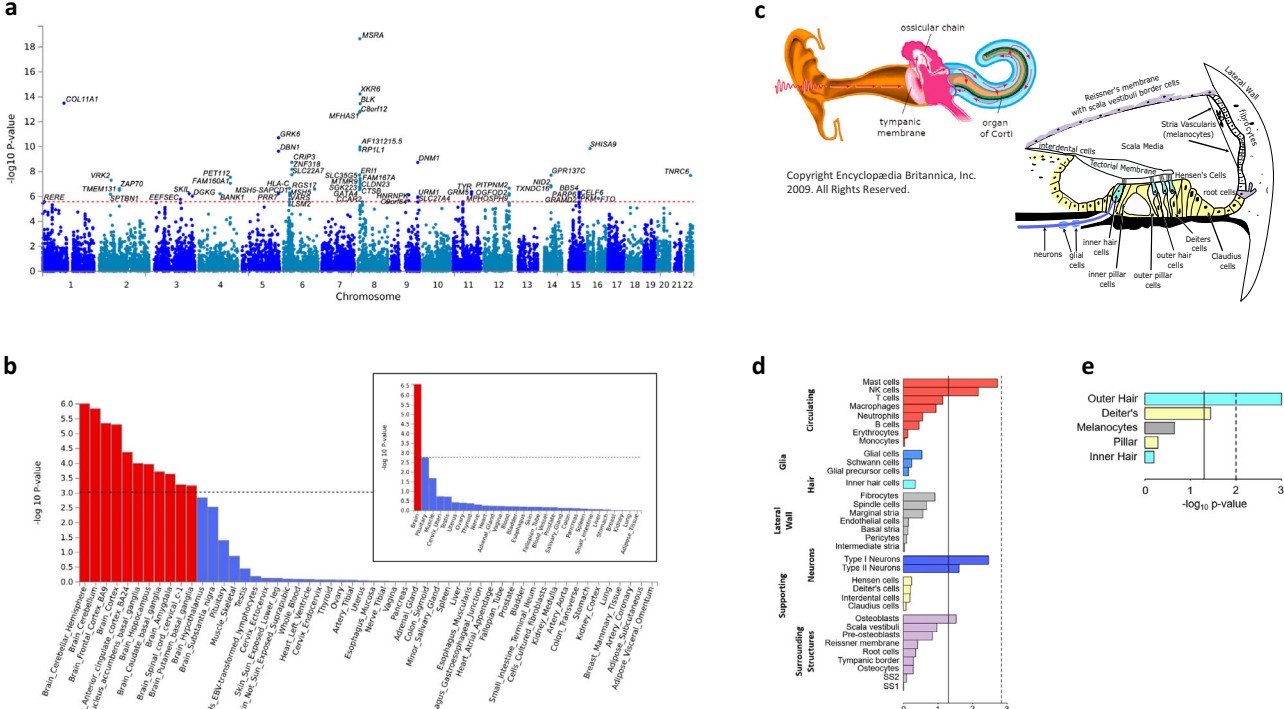

**Fig. 2 | Gene-based analyses of tinnitus identify 62 genes and gene expression enriched in brain tissue and specific ear cells. a** Manhattan plot of gene-based tinnitus meta-analysis across 481,874 subjects of European ancestry from MVP and UKB, showing 62 genome-wide significant genes. The $y$ axis represents -$\log_{10}$ $p$ values from the two-sided F-test for gene-based analysis. The red dotted line indicates the gene-wide significance threshold at $p < 2.66 \times 10^{-6}$ (Bonferroni correction for 18,972 genes tested). **b** MAGMA tissue expression analysis for gene expression of GTEx v8 data sets, showing significant enrichment in the brain (inlet), with 11 of 13 brain regions being enriched. To test for positive relationships between gene expression in a specific category and genetic associations, SNPs were mapped to 17,196 protein-coding genes and gene-property tests were performed for average gene-expression per tissue type conditioning on average expression across all tissue types. (Note: cerebellar hemisphere and cerebellum are the same tissue, with different RNA preservation after death). Bars denote -$\log_{10}$ $p$ values from one-sided t-tests. The dotted line indicates significance at $p < 9.26 \times 10^{-4}$ (Bonferroni correction for 54 specific tissues tested) and at $p < 1.67 \times 10^{-3}$ (Bonferroni correction for 30 general tissues tested) in the inlet. **c** Diagram of the ear, with red arrows showing direction of sound waves through the external ear, vibrating through the tympanic membrane and ossicles to the oval window into the inner ear. The inner ear is composed of three fluid-filled chambers, and the Organ of Corti rests atop the basilar membrane (black) in the center chamber, the scala media (orange in the upper figure). Organ of Corti is shown in cross-section below. Stria vascularis, the

locale for melanocytes, maintains the endocochlear potential required for outer hair cell (OHC) and inner hair cell (IHC) nerve firing. Deiter's cells support OHCs, and IHCs are surrounded medially by border cells and laterally by inner pillar cells. OHCs lengthen and shorten, acting as a dampener/amplifier of the fluid wave. IHCs provide mechanotransduction, sending a neural signal through cochlear nerve fibers to the brain. Diagram of the ear from Encyclopædia Britannica, Inc., copyright 2009. All rights reserved. **d** Cochlear cell type enrichment analyses based on mouse data (Jean et al. 2023). Clustered bar chart depicting results of conditional MAGMA gene-property analyses for 34 different cell types. Bars (denoting -$\log_{10}$ $p$-values from one-sided t-tests) are clustered and colored by the general cell type tested: circulating (red), glia (blue), hair (light blue), lateral wall (gray), neurons (royal blue), supporting (yellow), and surrounding structures (purple). Bars depict the cell type association conditioned on the average of all 34 cell types. Solid line denotes $p < 0.05$. Dotted line indicates significance after Bonferroni-adjustment for 34 comparisons ($p < 1.47 \times 10^{-3}$). **e** Organ of Corti cell type enrichment analyses based on mouse data (Hoa et al. 2020). Bar chart depicting results of conditional MAGMA gene-property analyses for five different cell types. Bars (denoting -$\log_{10}$ $p$-values from one-sided t-tests) are colored by the cell type tested: hair cells (blue), supporting cells (yellow), and melanocytes (gray). For each tissue type, the bar depicts MAGMA analysis conditioned on the average of the five cell types. Solid line denotes $p < 0.05$. Dotted line indicates significance after Bonferroni-adjustment for five comparisons ($p < 0.01$).

highlighted the differences between tinnitus and HD (Fig. 4). Whereas tinnitus is highly enriched in most brain tissues, and adjustment for HD did not change expression patterns significantly, HD shows very little enrichment in the brain. As discussed above, these results do not consider cochlear tissue expression, as it is not yet available for humans. Lacking likely relevant tissue, CC-GWAS results between tinnitus and HD show the largest differences in tissue expression in brain tissue, such as the cerebellum.

### Genetic overlap of tinnitus and other traits

Extending analyses beyond comparisons with HD, $r_g$ of tinnitus were estimated with a broad range of 789 phenotypes with GWAS publicly available from the Psychiatric Genomics Consortium (PGC) and Complex Trait Genetics Virtual Lab[55] (Supplementary Data 18). We found 103 significant correlations ($p < 6.34 \times 10^{-5}$), led by correlations with auditory phenotypes (e.g. no hearing difficulty, $r_g = -0.52$, SE = 0.03),

and including phenotypes from domains such as mortalities and comorbidities (long-standing illness, disability or infirmity, $r_g = 0.26$, SE = 0.03), activities & social interactions (no illness, injury, bereavement stress in last 2 years, $r_g = -0.29$, SE = 0.04), pain (multisite chronic pain, $r_g = 0.31$, SE = 0.03), medication use (no medication for pain relief, constipation, heartburn, $r_g = -0.26$, SE = 0.03), as well as mental functions (frequency of tiredness /lethargy in last 2 weeks, $r_g = 0.25$, SE = 0.03), and psychiatric disorders and traits (Major depressive disorder, $r_g = 0.21$, SE = 0.03) (Fig. 5a).

To further analyze the joint genetic architecture of tinnitus with psychiatric and health related traits we performed genomic structural equation modeling (gSEM)[56] including 24 phenotypes selected across the most significant domains (Supplementary Data 19). Genetic correlations across tinnitus, HD, and ten psychiatric disorders indicated strong clustering among the internalizing disorders and the hearing-related variables (Supplementary Fig. 4a).

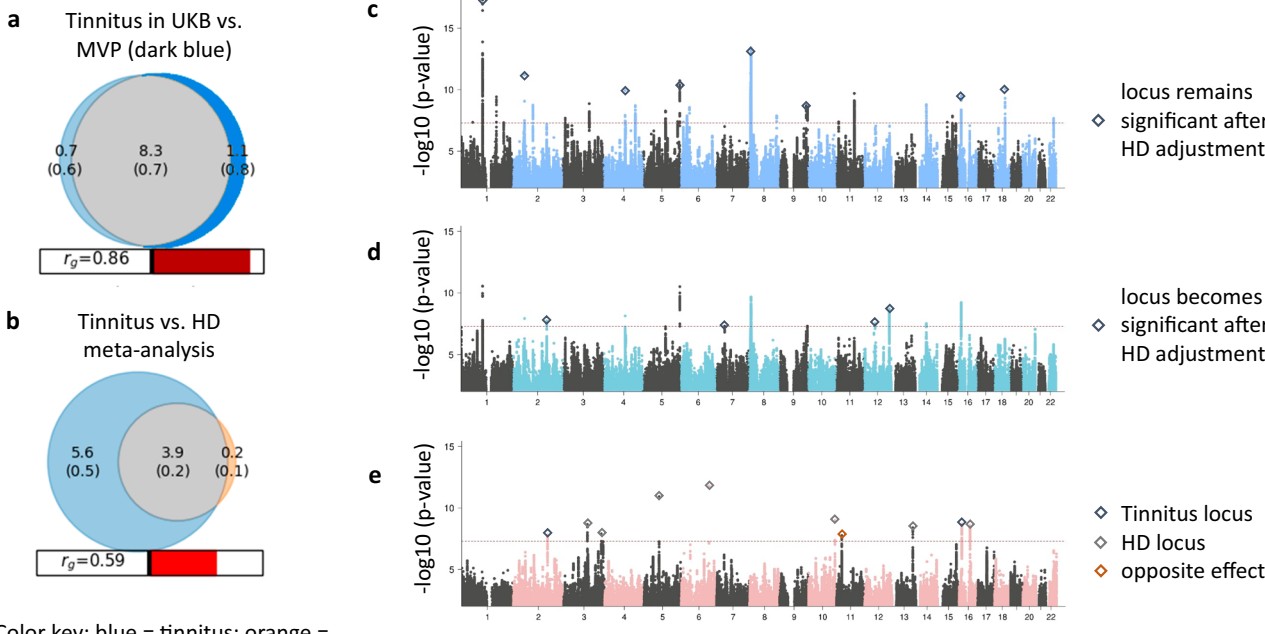

**Fig. 3 | Genetic architecture of tinnitus and its relationship to hearing difficulty (HD). a** Quantification of the polygenic overlap between tinnitus in the MVP and UKB. Bivariate MiXeR modeling estimates 8,262 shared causal variants between the cohorts (gray), accounting for the majority of variants influencing tinnitus (88.7% in MVP, 91.8% in UKB, respectively), with very little unique polygenic components (blue shades). **b** Polygenic overlap between tinnitus and hearing difficulty (HD), indicating an estimated 3,850 causal variants are shared, thus accounting for 95.4% of the variants influencing HD, but only 40.7% of the variants influencing tinnitus. The numbers (in thousands, with standard errors in parenthesis) in the Venn diagrams indicate the estimated quantity of causal variants per component, explaining 90% of SNP heritability for each phenotype. The size of the circles reflects the degree of polygenicity. Genetic correlations (rg) estimated between the two phenotypes are shown below the Venn diagrams. **c**–**e** Manhattan plots of tinnitus GWAS and different models incorporating hearing difficulty indicate unique and shared risk loci between the two phenotypes. The *y* axis represents -$\log_{10}$ p-values from two-sided z-tests. The red dotted line indicates the genome-wide significance threshold at $p < 5 \times 10^{-8}$. **c** GWAS meta-analysis of tinnitus, including participants from MVP ($N = 308,879$) and UKB (N = 172,995) identified 29 genome-wide significant (GWS) loci. The red line indicates the genome-wide significance threshold at $p < 5 \times 10^{-8}$. Diamonds indicate 8 tinnitus loci that will remain significant after adjustment for hearing difficulty (in **d**). **d** Tinnitus meta-analysis including a covariate for HD (based on ICD and self-report). Diamonds indicate 4 tinnitus loci that become significant after adjustment for HD. **e** Case-case GWAS of tinnitus and HD, showing 10 loci with significantly different effects between the two phenotypes. Blue diamonds indicate 2 loci with larger effects for tinnitus, green diamonds indicate 7 HD loci. The red diamond represents a locus with opposite effect (not GWS in either of the GWAS).

Genomic SEM factor loadings of >.25 from odd chromosome exploratory factor analysis (EFA) were used to inform two, three, and four-factor confirmatory factor analysis (CFA) models using even chromosomes (Supplementary Data 20A). The four-factor model provided the best fit ($\chi^2(48) = 340.325$, AIC = 400.326, CFI = 0.901, SRMR = 0.105) and identified a hearing factor with tinnitus and HD, a second factor with PTSD, ADHD, problematic alcohol use, and MDD, the third factor with MDD, Tourette's, anorexia, and OCD, and a fourth factor with schizophrenia, bipolar disorder, and autism spectrum disorder (Fig. 5b). Factor 1 explained 57% of the variance in tinnitus and HD. Correlations between factors ranged from $r = -0.09$ (0.05) to $r = 0.50$ (0.04). Latent factors 1 and 2 were positively correlated, suggesting that tinnitus and HD are modestly related to internalizing psychiatric conditions, but are also distinct from other types of psychopathology. Similarly, a gSEM analysis including tinnitus and 13 selected health-related traits was performed and the four-factor model provided the best fit ($\chi^2(70) = 434.2329$, AIC = 603.29, CFI = 0.986, SRMR = 0.071) (Supplementary Data 20B, Supplementary Fig. 4b). The four-factor model identified a "psychological distress" factor with tinnitus, neuroticism, and a negative loading for subjective wellbeing, a "physical distress" factor capturing overall poor health including short sleep duration, presence of illness or injury, pain, and tiredness, a "chronic illness" factor, and a "headache" factor including self-reported migraine and medication use including paracetamol (Fig. 5c). Correlations between factors ranged from $r = 0.52$ (0.10) to $r = 0.92$ (0.03). These findings suggest a meaningful relationship between tinnitus and psychological distress shared with subjective wellbeing and neuroticism, as well as moderate associations with other health factors, particularly physical distress, and headache.

## Discussion

Various theories regarding the generation and perception of tinnitus have focused on models of central neural gain following an initial injury to the cochlea. Decreased spontaneous firing rate from the cochlea following injury leads to central neural hyperactivity[57]. Successive nuclei along the auditory pathway amplify this spontaneous neural activity, as indicated in successive waves of the auditory brainstem response, possibly secondary to a mismatch between inhibitory (GABA) and excitatory (glutamate) neurotransmitter networks[58-60]. Neural firing pattern changes as well as increased neural synchrony has been noted in the auditory cortex following noise trauma[61]. Altered inhibitory neurotransmission in the auditory cortex has also been associated with tinnitus, and a reduction in GABA has been seen in the auditory cortex of tinnitus subjects[62].

For the first time we now have robust genetic data on close to 600,000 individuals supporting these theories. This study brings together many of the theoretical constructs surrounding the intracranial generation and perception of tinnitus, integrating them into the genetic domain. We identify 29 tinnitus loci in GWAS of EA, adding 26 loci to the 8 loci previously reported in the UKB[20-22]. Including Latinx and African ancestries in this first cross-ancestry GWAS for tinnitus adds 9 additional novel loci. Accordingly, genetic risk score predictions for tinnitus improved significantly with these new data. However, we currently still

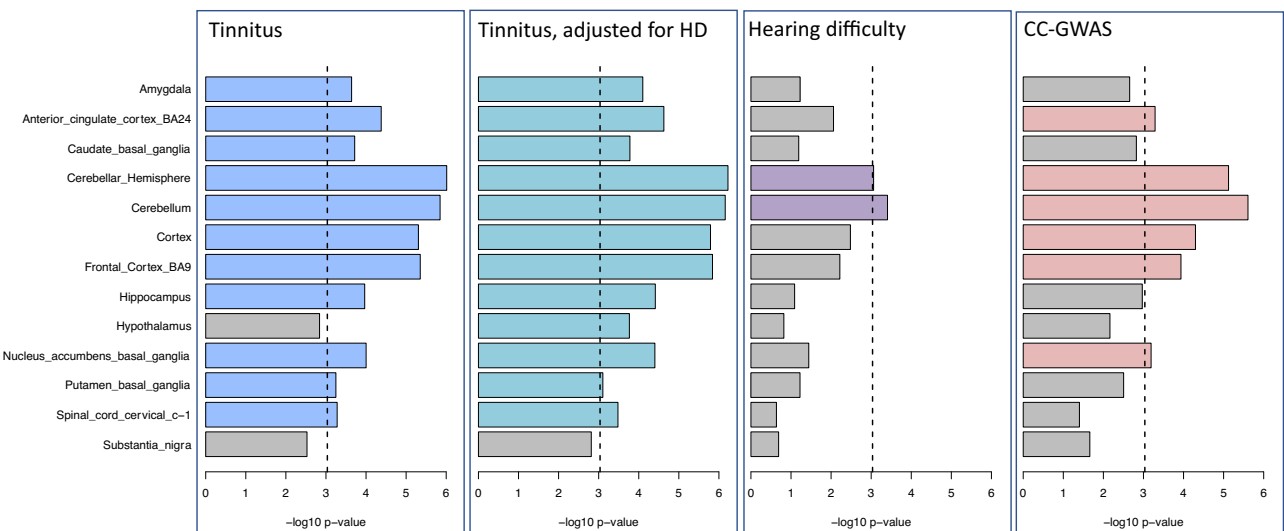

**Fig. 4 | Enrichment of specific brain regions in MAGMA tissue expression analyses, comparing GWAS on tinnitus with a GWAS on tinnitus adjusted for hearing difficulty, with a hearing difficulty GWAS, and a case-case GWAS on tinnitus versus hearing difficulty.** MAGMA tissue expression analysis for gene expression of GTEx v8 data sets, showing significant enrichment in specific brain regions for GWAS meta-analyses across the same set of MVP and UKB European ancestry participants ($N = 481{,}874$). To test for positive relationships between gene expression in a specific category and genetic associations, SNPs were mapped to 17,196 protein-coding genes and gene-property tests were performed for average gene-expression per tissue type conditioning on average expression across all tissue types. Bars denote -$\log_{10}$ p-values from one-sided t-tests. The dotted line represents Bonferroni-adjusted significance at $p < 9.26 \times 10^{-4}$ for 54 specific tissues (only brain tissues are shown here, none of the other tissues were significant).

predict only ~1.2% of the tinnitus variance in the EA population, and PRS performance is significantly lower for non-European subjects, as correlation between true and genetically predicted phenotypes decays with genetic divergence from the discovery GWAS[37]. Additional GWAS data on both European and non-European patients is needed to increase the number of SNPs that can explain the tinnitus heritability $h^2_{SNP}$ of 7%. Cohorts with a high prevalence of tinnitus, such as the MVP, may be particularly informative for gene discovery.

Gene-based analysis categorizes multiple genes expressed within the synaptic area. Although this finding affects all synapses, it is consistent with findings of deafferentation of the inner hair cell synapse as a suggested area of cochlear injury[63]. This loss of synapses, occurring in aging mice as well as noise levels that do not induce objective audiogram changes, can be associated with a reduced auditory brainstem response Wave I, an indication of diminished connections from the cochlear spiral ganglion to the dorsal cochlear nucleus (DCN)[64], however human studies have been problematic[65]. Specifically, while this cochlear injury has been suggested as an initial injury, it has yet to be linked to the onset of tinnitus.

One of these synaptic genes, *GRK6*, controls several GABA receptors[66], and associated with decreased stimulation of the DCN are tinnitus-related reductions in GABA levels in the auditory cortex[67]. This "release of inhibition" is indicated by increased spontaneous activity, bursting, enhanced sound-evoked response, and reduced neurochemical markers of inhibitory neurotransmission[68].

Our drug-class analysis indicates significant enrichment in targets of hypnotics, sedatives, muscle relaxants, and anxiolytics, all related to the expression of GABA Type A receptors. Restoration of the excitatory-inhibitory balance would be consistent with these drug class findings and provides a new area of pharmaceutical research for treatment. It is known that dopamine receptor knockout animals demonstrate increased vulnerability to acoustic injury[60,63,69]. Dopamine receptors are expressed on glutamatergic terminals in the inner hair synaptic area and spiral ganglion cells, where they modulate the excitatory glutamate response[70]. *GRK6* ($p < 1.92 \times 10^{-11}$) is a controller of dopamine sensitivity in both the brain and the cochlea[71].

While we confirm a high genetic correlation between the disorders ($r_g = 0.59$, SE $= 0.01$), emerging statistical techniques help delineate distinct differences in the genetic architecture of tinnitus and hearing loss. MiXeR estimates that the majority (95.4%) of the variants influencing HD are shared with tinnitus, while conversely, a separate 59.3% are unique to tinnitus. Tinnitus generally differentiates itself from HD by higher polygenicity and variants with lower discoverability. Since SNP identification is biased toward variants with larger effect sizes, findings are thus directed more toward hearing loss variants with higher discoverability. Future GWAS in larger cohorts will aid in identification of additional unique tinnitus risk loci.

Even with high genetic overlap, comparison of MAGMA tissue expression analyses shows a clear difference between hearing loss and tinnitus, with broad expression in multiple areas of the brain for tinnitus, versus HD which is confined to the cerebellum in our study. These results are consistent with the fact that tinnitus has its source of production and perception within the wider area of the brain, rather than the narrower focus of the cochlea. Broad changes in gray and white matter, magnetoencephalography indication of tinnitus-frequency specific activity, and a difference in the default mode network (DMN) of tinnitus subjects have been identified, while hearing loss appears to be related more specifically to the cochlea and auditory pathway[72–75]. However, the lack of human cochlear tissue in reference databases currently precludes a direct comparison across brain regions and cochlea. It must be added that severe tinnitus has been shown to be highly correlated with hyperacusis, i.e., increased sensitivity to loudness comfort level, and that would need to be addressed in future studies. (REF Cederroth's paper).

Both genetic correlations and gSEM demonstrate significant correlations between tinnitus and psychiatric and health-related traits, as supported by epidemiological studies[76,77]. The strongest associations are with self-reported hearing loss, speech understanding in noise, and hearing aid use. However, other findings of note include relationships with pain syndromes, wellness measures, and internalizing disorders. Tinnitus and another subjective symptom, pain, involve similar networks in the brain[78–80]. Findings associated with both pain and tinnitus

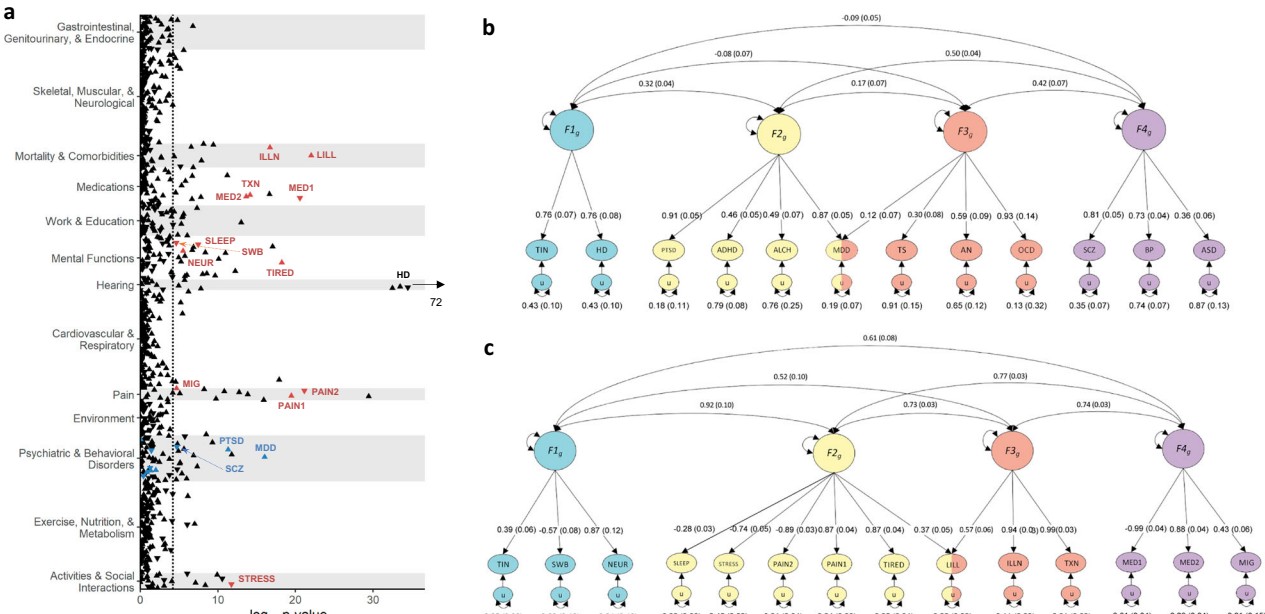

**Fig. 5 | Genetic correlations of tinnitus with psychiatric disorders and health related traits. a** Genetic correlations of tinnitus with 789 health-related traits and disorders across 13 domains. GWAS summary data was derived from the Psychiatric Genomics Consortium and the Complex Trait Genetics Virtual Lab (see Methods and Supplementary Data 18). The *y* axis represents -$\log_{10}$ p-values from the two-sided z-tests for genetic correlations. Orientation of the triangle indicates positive (up) or negative (down) correlations. Black arrow indicates that 3 hearing-related traits (from UKB) with p-values extending beyond the range of the X-axis ($p = 1.26 \times 10^{-72}$). Dotted line indicates Bonferroni-corrected significance at ($p < 6.34 \times 10^{-5}$). Selected psychiatric (blue, **b**) and health-related (red, **c**) traits were included in gSEM analyses. **b** Path diagram and standardized estimates from the best fitting confirmatory-factor model (CFA) of tinnitus, hearing difficulty, and 10 psychiatric disorders. Exploratory factor analyses of the genetic correlation matrix produced from multivariable LD-score regression of odd chromosomes were used to inform CFAs, fit to the covariance matrix from the even chromosomes. In this best-fitting CFA, four correlated latent genetic factors ($F1_g$, $F2_g$, $F3_g$, $F4_g$) represent shared genetic liability for the conditions. Single-headed arrows represent partial

regression coefficients and reflect the degree of relationship between the latent factor and each variable. Variation explained by latent factors can be computed by squaring the factor loadings. Curved, double-headed arrows represent correlations between factors. Unique variance not explained by the model in each condition is represented by the 'u' oval estimates. **c** Path diagram and standardized estimates from the best fitting CFA of tinnitus and 13 health-related traits. TIN tinnitus, HD hearing difficulty, PTSD post-traumatic stress disorder, ADHD attention-deficit/ hyperactivity disorder, ALCH problematic alcohol use, MDD major depressive disorder, TS Tourette's syndrome, AN anorexia nervosa, OCD obsessive compulsive disorder, SCZ schizophrenia, BP bipolar disorder, ASD autism spectrum disorder, SWB subjective well being, NEUR neuroticism, SLEEP sleep duration, STRESS No Illness/injury/bereavement stress in last 2 years, PAIN1 Neck or shoulder pain experienced in last month, PAIN2 no pain experienced in last month, TIRED frequency of tiredness/lethargy in last 2 weeks, LILL long-standing illness/disability/ infirmity, ILLN Number of self-reported non-cancer illnesses, TXN Number of treatments/medications taken, MED1 no medication for pain relief/constipation/ heartburn used, MED2 Paracetamol used, MIG migraine.

include loss of gray matter in the ventromedial prefrontal cortex and abnormal thalamocortical oscillations[81]. The DMN in the resting state shows increased connectivity between DMN and the fronto-limbic-striatal system in tinnitus, echoing increased connectivity between DMN and the nucleus accumbens in chronic pain. The frontostriatal system functions to update a signal deviation from the predicted environment, serving to update predictions and both tinnitus and pain have been described as a continuous prediction error[79].

The present study finds that tinnitus genes broadly encompass those of hearing loss. Tinnitus genes are over-expressed in the brain and cochlea, in agreement with extensive imaging, auditory brainstem responses, and epidemiologic data. We have been able to differentiate variants that may be either within a pathway that induces both hearing difficulties and tinnitus, or tinnitus specific. Further work will attempt to characterize tinnitus with various aspects of audiology available in the medical record. Evidence for the significant genetic correlation of tinnitus with specific psychiatric disorders and health-related traits provides a framework for further genomic research of tinnitus.

## Methods
### Phenotype
*UKB:* The UKB recruited 503,317 adults between 2006-2010 out of a population of 9.2 million men and women (5.45% recruitment rate) registered in the UK National Health Service within England, Scotland,

and Wales who were randomly invited to participate[1]. Participants signed electronic consents and answered questionnaires regarding demographics, lifestyles, and self-reported health conditions. 197,975 participants answered questions about tinnitus and hearing difficulties. Self-reported tinnitus (data field 4803) was assessed using the categorical question, "Do you get or have you had noises (such as ringing or buzzing) in your head or in one or both ears that lasts for more than five minutes at a time?"[20]. An ordinal definition was used based on tinnitus frequency, ranging from 1= No, never, 2= Not now, but have in the past, 3=Some of the time, 4 = A lot of the time, to 5 = Most or all of the time (Supplementary Data 1A). Self-reported hearing difficulty (data field 2247) was assessed using the hearing question "Do you have any difficulty with your hearing?". Hearing difficulty was coded as yes (including completely deaf) or no.

*MVP:* The MVP made available data from 462,335 participants recruited since 2011 in version 18_2 (released March 27, 2019), which contains MVP enrollees through Jan 9, 2019, with MVP survey data through Jan 18, 2019 and ICD data from the Corporate Data Warehouse through September 30, 2018[82]. Participants filled out a basic health question survey, and information including ICD diagnostic codes has been linked to individual, de-identified health records. All participants provided written informed consent to participate. We assessed tinnitus using self-report data and ICD codes (Supplementary Data 1B). Cases included self-reported tinnitus on the baseline survey defined as those

who checked the box to the query: "Please tell us if you have been diagnosed with the following conditions: Tinnitus or ringing in the ears" and had evidence of answering any of the previous or subsequent questions (i.e., they did not miss the page), and those with an ICD diagnosis of tinnitus (H93.1X, 388.30, or 388.31) in the electronic health record (EHR), non-pulsatile. Controls were those who did not check the box for tinnitus (but had answered any of the previous or subsequent questions on the baseline survey) and did not have an ICD code for tinnitus. Self-reported hearing difficulty (HD) was assessed on the baseline survey as a checkmark in a box to the query: "Please tell us if you have been diagnosed with the following conditions: Severe hearing loss or partial deafness in one or both ears." Cases were those who checked the box and had evidence of answering any of the previous or subsequent questions (i.e., they did not miss the page), and controls were those were who did not check the box for hearing loss (but had answered any of the previous or subsequent questions on the baseline survey). The study was approved by the University of California San Diego and the VA CIRB and VASDHS R&D Institutional Review Boards.

### Genotyping, QC, and imputation
*UKB:* Analyses were based on the version 3 release of the UKB imputed genetic dataset. Details of genotyping, quality control, and imputation have been previously reported[83]. In brief, subjects were genotyped on either the Affymetrix Axiom or UK BiLEVE Axiom arrays, with approximately 4,700 samples per batch. Genotype calling was performed using a custom pipeline designed for biobank scale data. Quality control of marker genotypes included tests for batch and plate effects, deviations from Hardy-Weinberg equilibrium based on exact tests, sex effects, array effects, and discordance among technical replicates. Based on markers passing QC, subjects were removed for >2% missing genotype rate, discrepancy between self-reported sex and genetically determined sex, or excessive heterozygosity. Phasing was performed using SHAPEIT3[84] in partially overlapping chunks of 15,000 markers, with the 1000 Genomes Project Phase 3 (1KGPp3) dataset used as a reference panel. Chunks were merged using hapfuse. Data was imputed using IMPUTE[84,85] using the combination of the 1KGPp3, UK10K, and Haplotype Reference Consortium panels, where the latter was preferentially used as the imputation reference.

*MVP:* Analyses in this investigation were based on release 3 of the MVP imputed genotype data. Details of the genotyping, quality control, and imputation procedures used have been reported in detail[82]. In brief, MVP samples were genotyped on a customized version of the Affymetrix Axiom biobank array and standard genotype quality control procedures were followed. Genotype data was phased using Eagle version 2.4[86] and imputed using Minimac version 4[87] with the 1KGPp3 (version 5)[88] reference panel.

### Assessment of ancestry
*UKB:* ancestry was estimated using a standardized pipeline based on SNPweights[89] of 2,027 ancestry informative markers (https://github.com/nievergeltlab/global_ancestry)[90]. Subjects were classified into European ancestry (EA) if they had >90% EA proportion; all other subjects were excluded from analyses.

*MVP:* HARE (harmonized ancestry and race/ethnicity) estimates[91] were used to define subjects as non-Hispanic white (corresponding to European ancestry; EA), non-Hispanic black (corresponding to African ancestry; AA), and Hispanics (corresponding to Latinx; LAT) for GWAS. Relatedness was estimated using KING[92]. For each pair of subjects with an estimated kinship coefficient >0.0884 (2nd degree or closer), one individual was removed, with the preference to retain cases. If individuals had the same diagnostic status, one individual was removed at random.

### Calculation of principal components (PCs)
*UKB:* SNPs were excluded that had a minor allele frequency (MAF) < 5%, HWE $p > 1 \times 10$-3, call rate <98%, were ambiguous (A/T, G/C), located in the MHC region (chr6, 25-35 MB) or chromosome 8 inversion (chr8, 7-13 MB). SNPs were pairwise LD-pruned (r2 > 0.2) and a random set of 100 K markers was used for each subset to calculate PCs based on the smartPCA algorithm in EIGENSTRAT[93].

*MVP:* PCs were calculated within unrelated subjects of the same ancestry using FlashPCA2[94]. SNPs were excluded for MAF < 5%, HWE $p > 1 \times 10$-3, call rate <98%, were ambiguous (A/T, G/C), being located in the MHC region (chr6, 25-35 megabase (MB)) or chromosome 8 inversion (chr8, 7-13 MB). Remaining SNPs were pruned for LD over a 1 MB window stepped over 50 variants at a time with an $r^2$ threshold of 0.05. PCs were calculated in the pruned marker set.

### GWAS
*UKB:* GWAS was performed in Bolt LMM 2.3.2[95], using linear mixed models to account for relatedness, including the first 6 principal components, assessment center, and genotyping batch as covariates.

*MVP:* GWAS was performed separately for each of the 3 HARE groups in PLINK[96], using logistic regression including 10 PCs as covariates.

Analyses adjusted for hearing difficulty were conducted similarly, with an additional covariate included for hearing difficulty (based on self-report and ICD codes for the MVP to match the tinnitus definition).

### Meta-analysis
Sample size weighted fixed effects meta-analyses of MVP and UKB EA datasets, AA and LAT datasets, and cross-ancestry GWAS (including UKB EA and MVP EA, AA, and LAT) were performed in METAL[97], including SNPs present in all datasets. Effective sample size (4 / 1/(N cases) + 1/(N controls)) was used as the study weight for MVP, with observed sample size used as the study weight for the UKB. SNPs with MAF < 1% or imputation information score <0.6 in either cohort were excluded from meta-analysis.

Self-reported hearing difficulty (HD) summary statistics were included for comparison, based on a meta-analysis of a GWAS including 87,056 cases and 163,333 controls from the UKB[36] and a GWAS of 88,782 EA cases and 151,291 controls from the MVP (this study), following the methods described above for MVP.

### Chromosome 8p23.1 inversion analysis
Inversion genotype was determined using the invClust[98] R package. MVP and UKB EA participants were classified into homozygous inversion carriers, homozygous non carriers, or heterozygotes. To determine if tinnitus status was associated with a particular SNP in the inversion region, association analyses were performed stratified by inversion status, then meta-analyzed across strata. To determine if tinnitus was associated with the inversion itself, logistic regression of tinnitus on inversion status was performed.

### Regional association plots
Regional visualizations of genome-wide significant loci were produced using LocusZoom 1.4[99] LD was calculated using 1KGPp3 data, where EUR samples were used as reference genotypes for EA samples, AMR used for LAT, and AFR used for AA samples.

### Functional mapping and annotation
Functional annotation of GWAS results was performed with the FUMA pipeline version v1.3.7[100]. Annotations are based on human genome assembly GRCh37 (hg19). FUMA was used with default settings unless stated otherwise. The SNP2Gene module was used to define independent genomic risk loci and variants in LD with lead SNPs ($r^2$ > 0.6, calculated using ancestry-appropriate 1KGPp3 reference genotypes: EUR were used as reference genotypes for EA, AMR used for LAT, and AFR used for AA samples). SNPs in risk loci were mapped to protein-coding genes with a 10 kb window.

Functional consequences of SNPs were obtained by mapping the SNPs on their chromosomal position and reference alleles to databases containing known functional annotations, including ANNOVAR, Combined Annotation Dependent Depletion (CADD), RegulomeDB (RDB), and chromatin states in brain tissues/cell types. Next eQTL mapping was performed on significant (FDR < 0.05) SNP-gene pairs, mapping to GTEx v8 brain tissue, RNAseq data from the CommonMind Consortium and the BRAINEAC database. Chromatin interaction mapping was performed using chromatin interaction data from Giusti-Rodriguez et al.[101], including the dorsolateral prefrontal cortex, hippocampus and neuronal progenitor cell line and adult and fetal cortex tissue, and PsychENCODE data including Hi-C derived one way enhancer-promoter links and promoter anchored loops. An FDR < 1 × 10$^{-5}$ defined significant interactions, based on previous recommendations, modified to account for the differences in cell lines used here.

PheWAS of leading SNPs from risk loci was performed using the GWAS Catalog version e104_r2021-09-15[102] implemented in FUMA.

### Fine-mapping

Polygenic functionally-informed fine-mapping (Polyfun)[103] software was used to annotate our results data with per-SNP heritabilities derived from a meta-analysis of 15 UK Biobank traits, as performed in Weissbrod et al.[103]. Tinnitus risk loci were fine-mapped using SUSIE[104], with the maximum number of causal SNPs set to 2, per SNP heritabilities used as priors, pre-computed UKB based summary LD information used as the LD reference, and loci start and end positions as determined by FUMA.

### Polygenic Risk Score (PRS) analysis

Tinnitus GWAS summary statistics for MVP EA were used to calculate PRS for the UKB sample and vice versa. In addition, PRS based on the full meta-analysis were tested in a non-overlapping subset of MVP participants of European ancestry, including N = 36,921 tinnitus cases and 63,507 controls (phenotype defined as described above), using imputed genotype data from MVP release 4. GWAS summary statistics were filtered to common (MAF > =1%), well imputed variants (INFO > = 0.8) and Indels and ambiguous SNPs were removed. PRS-CS[35] was used to infer posterior effect sizes of SNPs, using the 1000Genomes Phase 3 EUR based LD reference panel supplied with the program, with the global shrinkage parameter set to 0.01, 1,000 MCMC iterations with 500 burn-in iterations, and the Markov chain thinning factor set to 5. PRS were calculated using the -- score option in PLINK 1.9, using the best-guess genotype data of target samples. For each SNP the risk score was estimated as the posterior effect size multiplied by number of copies of the risk allele. PRS was estimated as the sum of risk scores over all SNPs. PRS were tested for association with tinnitus in logistic regression analysis adjusted for principal components. To aid in interpretability of effect sizes, PRS were centered to the mean PRS and rescaled to have unit variance. The proportion of variance explained by PRS for each study was estimated as the difference in Nagelkerke's R$^2$ between a model including PRS plus covariates and a model with only covariates. R$^2$ was converted to the liability scale using standard formulae[105], assuming 12.5% population prevalence and the observed sample prevalence of 30%.

### Gene-based and gene set, and tissue-enrichment analyses with MAGMA

The MAGMA[106] tool implemented in FUMA was used to perform gene-based, gene-pathway, and tissue enrichment analyses. For gene-based analysis, SNPs were mapped to 18,873 protein coding genes. For each gene, its association with tinnitus was determined as the weighted mean squared test statistic of SNPs mapped to the gene, where LD patterns were calculated using ancestry appropriate 1KGp3 EUR reference genotypes. Significance of genes was set at a Bonferroni-corrected threshold of p = 0.05/18,873 = 2.65 × 10$^{-6}$. To see if specific

biological pathways were implicated in tinnitus, gene-based test statistics were used to perform a competitive set-based analysis of 15,485 pre-defined curated gene sets and GO terms obtained from MsigDB. Significance of pathways was set at a Bonferroni-corrected threshold of p = 0.05/15,485 = 3.2 × 10$^{-6}$. To test if tissue-specific gene expression was associated with tinnitus, gene set-based analysis was also used with expression data from GTEx v8 RNA-seq and BrainSpan RNA-seq, where the expression of genes within specific tissues were used to define the gene properties used in the gene-set analysis model.

### Cochlear cell type enrichment analyses

Gene-set analyses were performed using cochlear-cell types' gene expression values from Jean et al.[39] (postnatal day 20 mice only) and Hoa et al.[40]. Additional pre-processing steps were applied to data from Jean et al.: data was imported into the R package Seurat[107] version 4.9.9.9058 using function 'CreateSeuratObject()'. Nuclei with the following parameters were kept for further analysis: RNA count between 1,000 and 30,000; gene/non-coding RNA number between 500 and 5,000 and the percentage of mitochondrial genes less than 2%. 38,144 cells remained after quality control. Pseudobulk gene expression from each cochlear cell type were calculated as the mean of raw expression level across all the cells in each cell type and log2 normalized. Data from Hoa et al. for the 5 tissues from the organ of Corti had already been pre-processed into reads per kilobase million by study authors[40]. To convert mouse gene ensemble IDs to human gene entrez IDs, a homology map was obtained from the Mouse Genome Database[108]. Genes with duplicated identifiers were removed. To perform the MAGMA gene-set analyses, the --gene-covar flag was used, to perform the following linear regression model:

$$Z = \beta_0 + X\beta_s + C\beta_c + \varepsilon \qquad (1)$$

where Z is the Z-score of the gene-level association statistic, $\beta_0$ is an intercept term, X is the continuous measure of gene expression, C is a matrix of covariates including gene length, correlation between genes based on LD computed in the gene-level association analysis, and average of gene expression taken across all cell types analyzed, and $\varepsilon$ is a term for random error. As a complementary analysis, LDSC cell type specific analyses were conducted with the --h2-cts flag in LDSC. Per Finucane et al.[109], for each focal tissue, genes were ranked based on their t-statistic for cell type specific expression, the top 10% of genes were selected, and a 100-kb window was used to obtain a genome annotation. Bonferroni multiple-testing correction for the number of cell types and tissues was applied for all tests performed.

### Drug-class and drug-set enrichment analyses

Drug set and drug class analyses were performed as described previously[110]. MAGMA[106] competitive gene-set analyses were restricted to drug sets (genes targeted by individual drugs). Gene boundaries were extended 35 kb upstream and 10 kb downstream from the boundaries in build 37 reference data from the NCBI, to include regulatory regions outside the transcribed region. Gene-level association statistics were defined as the aggregate of the mean and the lowest variant-level p-value within the gene boundary, converted to a Z-value. Drug sets comprised the targets of each drug in the Drug–Gene Interaction database DGIdb v.4.2.0[42], the Psychoactive Drug Screening Database Ki DB[43], ChEMBL v27[44], the Target Central Resource Database v6.7.0[45], and DSigDB v1.0[46] (all downloaded in October 2020).

Results from the drug set analysis were used to assess the enrichment of tinnitus genetic signal in drug classes, defined according to the Anatomical Therapeutic Chemical class of the drug[110]. We analyzed only drug classes with at least 10 valid drug gene sets within them. Enrichment was quantified as the area under the receiver operating characteristic curve of the drug sets ranked by their association in the drug set analysis. For a given drug class, an enrichment curve was drawn

scoring a "hit" if the drug gene set was within the class, or a "miss" if it was outside of the class. Enrichment was tested via Wilcoxon Mann-Whitney tests comparing drug sets within the class to those outside of the class[110]. Multiple testing was controlled using Bonferroni correction ($p < 3.26 \times 10^{-5}$ for drug-set analysis and $p < 3.13 \times 10^{-4}$ for drug-class analysis, accounting for 1,534 drug-sets and 160 drug-classes tested).

### SNP-based Heritability and genetic correlation
SNP-based heritability ($h^2_{SNP}$) and genetic correlations ($r_g$) were evaluated using LD score regression[111]. Input LD scores were computed from 1KGPp3 EUR samples. The LDSC intercept ($h^2_{INT}$) was used to test for artifactual inflation of test statistics and the attenuation factor (($h^2_{INT} -1$)/(mean($\chi^2$)−1) was used to estimate the proportion of inflation coming from polygenic signal. The $h^2_{SNP}$ of MVP tinnitus was converted to the liability scale using standard formula[105]

$$h^2_{liab} = h^2_{obs} \frac{K^2(1-K)^2}{P(1-P)z^2} \tag{2}$$

where $h^2_{liab}$ is the liability scale SNP based heritability, $h^2_{obs}$ is the observed scale heritability, $K$ is the population prevalence, $P$ is the sample prevalence, and z is the normal distribution density function evaluated at the normal quantile function evaluated at $K$. Tinnitus and HD GWAS summary statistics were uploaded to the Complex Trait Genetics Virtual Lab[55] (https://vl.genoma.io/). Cross-trait LDSC[111] was performed with all phenotypes in the database with $h^2_{SNP}$ z-score > 4 ($N = 772$), excluding tinnitus and HD-related phenotypes. Bonferroni adjustment was made for 772 comparisons ($p \le 6.48 \times 10^{-5}$). To identify genetic differences between tinnitus and hearing, $r_g$s were contrasted using z-tests.

### Univariate and bi-variate Gaussian mixer model (MiXeR) analysis
We used univariate MiXeR v1.3[112] to estimate the genetic architecture of phenotypes. MiXeR estimates SNP-based heritability and two sub-components whose product is proportional to heritability: the proportion of non-null SNPs (polygenicity) and variance of effect sizes of non-null SNPs (discoverability). MiXeR was applied to GWAS summary statistics under the default settings with the supplied EA LD reference panel. The results reported for the number of influential variants reflects the number of SNPs necessary to explain 90% of SNP based heritability. Bivariate MiXeR[113] was used to estimate phenotype specific polygenicity and the shared polygenicity between phenotypes. Goodness of fit of the MiXeR model relative to simpler models of polygenic overlap was assessed using AIC values. Heritability, polygenicity and discoverability estimates were contrasted between datasets using the z-test.

### Case-Case GWAS
Case-Case GWAS was performed using the CC-GWAS method[53]. Tinnitus and hearing difficulty meta-analyses were supplied as program inputs. As program input parameters, population prevalences of tinnitus and hearing difficulty were both set to 12.5%, with lower and upper bounds for prevalences respectively set to 5% to 50%, the liability scale heritability of tinnitus was set to 7% and hearing difficulty to 9% (empirical estimates of $h^2_{SNP}$ from this data), genetic correlation set to the empirical estimate of 0.57, a genetic covariance intercept of 0.22, and the number of effective loci set to 10,000.

### Cross-trait genetic correlations
Cross-trait LDSC[111] was performed with the EA tinnitus GWAS meta-analysis summary statistics and summary statistics of 10 psychiatric disorders from the Psychiatric Genomics Consortium (PGC). In addition, tinnitus summary statistics were uploaded to the Complex Trait Genetics Virtual Lab[55] (https://vl.genoma.io/) and cross-trait LDSC was performed with all phenotypes in the database with $h^2_{SNP}$ z-score > 4

($N = 779$ phenotypes). Genetic correlations ($r_g$) were considered significant at $p < 6.34 \times 10^{-5}$ (Bonferroni corrected for 789 comparisons). For each trait, domain name and chapter was assigned based on information from the GWAS atlas[100,114].

### Genomic structural equation modeling (gSEM)
Genomic structural equation modeling was carried out with R package GenomicSEM[56]. Multivariable LD score regression[111] using 1KGPp3 EUR reference was used to estimate the genetic covariance matrix (S), and corresponding sampling covariance matrix (V) for tinnitus, hearing difficulty, and psychiatric disorders and health related traits (Supplementary Data 19). Quality control consisted of removing SNPs with a MAF < 0.01 and MHC and filtering to Hapmap3. Exploratory factor analysis (EFA) was conducted on the odd chromosomes using R factanal function. EFA was used to estimate the appropriate number of latent factors in the model. Confirmatory factor analyses (CFAs), informed by EFA, were fit to the covariance matrix from the even chromosomes. A separate set of chromosomes were used for EFA and CFA to avoid overfitting. Traits with standardized EFA factor loadings exceeding .35 were assigned to factors in the CFA. For some EFA solutions, traits not reaching the .35 criteria for any factor, were assigned using a more lenient threshold of .20. When a factor had only two traits, loadings were set to equal to maintain identification. CFAs were fit using the weighted least squares (WLS) estimator, and model fit was evaluated using p-value for the chi-square test, Akaike Information Criterion (AIC), Comparative Fit Index (CFI), and Standardized Root Mean Square Residual (SRMR). Standardized loading values are reported.

### Reporting summary
Further information on research design is available in the Nature Portfolio Reporting Summary linked to this article.

## Data availability
Summary statistics for the UKB GWAS used in this study have been deposited on figshare repository (https://doi.org/10.6084/m9.figshare.24121281.v1)[115]. Summary statistics are publicly accessible on figshare; raw data are protected and are not available due to privacy reasons. Summary statistics for MVP analyses will be deposited upon publication on dbGaP under accession number phs001672 (https://www.ncbi.nlm.nih.gov/projects/gap/cgi-bin/study.cgi?study_id=phs001672.v11.p1). MVP summary data access can be obtained by submitting a data access request through dbGaP; raw data are protected and are not available due to privacy reasons. The dataset from Hoa et al. was made available through correspondence with the authors, and the dataset from Jean et al. are available on the gEAR portal (https://umgear.org/index.html?share_id=9c42d685&gene_symbol_exact_match=1). The programs LocusZoom, Polyfun, and FUMA provide the reference panels and datasets used in the described analysis; drug-class and drug-set analyses were done using the Drug Gene Interaction Database DGIdb v4.2.0 (https://www.dgidb.org/downloads), Psychoactive Drug Screening Database Ki Database (https://pdsp.unc.edu/databases/kiDownload/), ChEMBL v27 (https://chembl.gitbook.io/chembl-interface-documentation/downloads), Target Central Resource Database v6.7.0 (http://juniper.health.unm.edu/tcrd/download/), and DSigDB v1.0 (https://dsigdb.tanlab.org/DSigDBv1.0/download.html).

## Code availability
The codes for the analysis are available on Github (https://github.com/nievergeltlab/Tinnitus)[116].

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

## Acknowledgements

This research is based on data from the Million Veteran Program, Office of Research and Development, Veterans Health Administration, and was supported by awards RX002744-01 to A.F.R., R.E.C., and C.M.N.; BX005920-01 to C.M.N. and R.E.C.; BX001205 to A.F.R; and RX004293-01 to R.E.C., C.M.N., and A.F.R. from the VA San Diego Healthcare System and by the VA San Diego Healthcare System Center of Excellence for Stress and Mental Health. This publication does not represent the views of the Department of Veteran Affairs or the United States Government. Role of the Funder/Sponsor: The sponsors had no role in the design and conduct of the study; collection, management, analysis, and interpretation of the data; preparation, review, or approval of the manuscript; and decision to submit the manuscript for publication. Additional Contributions: Research was performed using the UK Biobank Resource under application 40951. We thank all the participants in the UK Biobank and the VA Million Veteran Program.

## Author contributions

C.M.N., R.E.C., and A.X.M. contributed equally to this work as co–first authors and last author. Obtained funding for studies: C.M.N., R.E.C, A.F.R. Clinical: R.E.C. Statistical analysis: A.X.M, C.C., C.T., K.H., E.A.M., F.T., M.G., N.P.D., J.R.I.C., Y.Z. Writing group: A.F.R., A.X.M., C.M.N., J.R.I.C., M.B.S., M.G., N.P.D., R.E.C.

## Competing interests

M.B.S. has in the past 3 years received consulting income from Acadia Pharmaceuticals, Aptinyx, atai Life Sciences, BigHealth, Bionomics, BioXcel Therapeutics, Boehringer Ingelheim, Clexio, Eisai, EmpowerPharm, Engrail Therapeutics, Janssen, Jazz Pharmaceuticals, NeuroTrauma Sciences, PureTech Health, Sumitomo Pharma, and Roche/Genentech. M.B.S. has stock options in Oxeia Biopharmaceuticals and EpiVario. He has been paid for his editorial work on *Depression and Anxiety* (Editor-in-Chief), *Biological* Psychiatry (Deputy Editor), and *UpToDate* (Co-Editor-in-Chief for Psychiatry). A.F.R. is a co-founder of and holds stock in Otonomy, Inc. All other authors declare no competing interests.
