## [Peer Review File · Nature Communications]

Genetic architecture distinguishes tinnitus from hearing lossREVIEWER COMMENTS

Reviewer #1 (Remarks to the Author):

Review

Clifford et al. follow-up on their previous GWAS study to better understand the differences between hearing loss and tinnitus from a genetic point of view. The authors perform a meta-analysis of the former datasets, and include non-European ancestries, to identify 39 loci. With enrichment analyses, they show that tinnitus finds multiple genes located in the brain, unlike hearing loss. They use single cell RNA sequencing data of the mouse adult cochlea to identify the peripheral contributors to tinnitus.

I congratulate the authors for this massive effort! This manuscript addresses a major issue in the field, and provides a significant advance. From a genetic point of view, it is seldom seen that GWAS try to disentangle two co-occurring conditions that are so closely related. Thus, this study will also provide important insights to the genetic community at large. The paper is very well written (with the exception of the introduction, half of which are results), follows a logical analytical flow, the majority of the statistics are sound. The scRNAseq maybe deserves some additional validation in order to exclude the possibility of false positives. This would require the analysis of additional cellular datasets. Finally, the summary statistics from the Million Veterans will be made available after publication – while that of the UKBB is accessible to the biobank itself: this needs to be addressed as for the sake of progress in science, and the needs of the reviewers to verify the claims, the data needs to be available now. With hopes these aspects can be addressed, then the manuscript would reach the expected quality.

Major comments:

1. The scRNAseq analysis uses a single data set by Hoa et al. This analysis is conditioned on liver, which is a novel way to address and improve the analysis. However, this protocol remains to become a standard. The authors should include the following datasets on separate confirmatory analyses. The analyses should include the traditional LDSC and MAGNA analyses separately.

a. Jean et al. Proc Natl Acad Sci U S A. 2023 Jun 27;120(26):e2221744120. doi: 10.1073/pnas.2221744120. Epub

b. Milon et al. Cell Rep. 2021 Sep 28;36(13):109758. doi: 10.1016/j.celrep.2021.109758.

c. Sun et al. Protein Cell. 2023 Apr 13;14(3):180-201. doi: 10.1093/procel/pwac058.

d. Xu et al. Front Cell Neurosci. 2022 Aug 18;16:962106. doi: 10.3389/fncel.2022.962106. eCollection 2022.

e. Iyer et al. Elife 2022 Nov 29;11:e79712. doi: 10.7554/eLife.79712.

2. Line 318 and subsequent paragraph: authors refer to the synaptopathy from Liberman and the article from Schaette. The manuscript from Liberman performed in mice has been cited nearly 1000 times but never effectively reproduced. Likewise, the paper from Schaette has not yet been properly validated. I would minimize all references to synaptopathy and hidden hearing loss.

3. Authors should make the UKBB and MVP summary stats available now with a doi number accessible for the reviewers and released upon publication (e.g. zenodo). The GWAS data from both the UKBB and the MVP published in JAMA Otolaryngology Head and Neck Surgery was not accessible anywhere. This is quite a shame. The codes of the analysis should also be made available. I understand this is a major effort, but the auditory neuroscience community, which is not specialist in the field of genetics, would deserve having such methodologies available to advance the science on other fronts.

Minor comments:

- The abstract does not convey the major findings from the paper, and ends blunt on a broad analytical statement. More of the major results should be reported here (what brain regions was tinnitus enriched... etc ..), with conclusions and perspectives on the knowledge the work has generated.
- Half of the introduction, starting from the third paragraph, summarizes the results. Authors should keep the introduction to present the background knowledge in the field rather than stating what they found. Delete the whole paragraph (can be moved to the end of the discussion/conclusions) and include additional information on tinnitus and hearing loss, both environmental and genetic influencers.
- The first paragraph of the intro should mention the two studies that have shown that tinnitus is associated with increased latencies of the ABR Wave V (inferior colliculus). These studies are important because they are mutually confirmatory, despite the different analytical approaches (one stratifies by hyperacusis, the other one includes it as a variable in a statistical model)
 - o Hofmeier et al. Clin Transl Med. 2021 May;11(5):e378. doi: 10.1002/ctm2.378.
 - o Edvall et al. J Clin Invest. 2022 Mar 1;132(5):e155094. doi: 10.1172/JCI155094.
- Some of the references are wrong:
 - o Line 51: Ref 13 is on Alzheimer's risk, not on loss of inhibition
 - o Line 52: Ref 16 is an adoption study, not a twin study
 - o Line 59: The first study from Amanat was predominantly Spanish, and replicated with a smaller set of a Swedish population.
- Supplementary table 1 should present the numbers for the meta-analysis
- The very high prevalence of tinnitus in the MVP likely accounts for the high power of this analysis. Somewhere, the reader should understand that this is a very untypical cohort to work with.
- Typo on Sup Table 16 (Adjutyed)
- Line 348: magnetoencephalography is a method – not a brain change.
- Paragraph starting line 352: Please refer only to causal inferences (longitudinal or case-control), not cross-sectional. Discuss both the epidemiology of hearing loss and tinnitus.

Reviewer #2 (Remarks to the Author):

The manuscript by Clifford et al describes a GWAS tinnitus performed in the VA and UKBB study sets. The phenotyping is based on self-reported questionnaires and/or ICD codes. The study samples are large and provide a very nice opportunity to tackle the genetic background of tinnitus.

As many other complex diseases, tinnitus is not an independent disease but a symptom that associates with many other diseases/symptoms including hearing loss, depression and musculoskeletal traits. Thus, the dissection of "tinnitus specific gene loci" is challenging.

The manuscript includes comprehensive analyses. It also is, for the most part, clearly written. However, some parts of the presentation in the main manuscript could be improved. Specifically, this reviewer would like to have a clearer presentation of loci that are "tinnitus specific/enriched" and their pleiotropy. Now most of the downstream analyses include hearing loss variants, which is logical due to the larger number of associated variants compared to "tinnitus specific loci". However, this potentially also misses some interesting aspects.

Figure 3 presents the analytical attempt to distinguish between tinnitus and HD loci. This reviewer finds this part of the paper the most interesting one. Yet, reading the results is not easy. While Manhattan plots are visually easy to grasp, they are not very informative in this setting. A table would be a better format. Now the actual results and the table is in the Supplement. This reviewer would recommend a simplified table produced from Supplementary tables 16 and 17. The table could also include the nearest gene/genes around the lead variant and pleiotropy analyses of the lead variant.

Also, a pleiotropy table of the main Table 1 as a Supplementary table might be helpful. The concern of this reviewer is that the current chapter "Genetic overlap of tinnitus and other traits" has a potential bias towards mental health due to the resources used for the analysis (especially the PGC dataset). This might miss some of the potential genetic correlations with autoimmune diseases, skeletal disorders and hypothyroidism. Was a locus specific PheWAS analysis performed using VA and UKBB datasets?

In summary from the above, the current form of the presentations does not quite address the title "Genetic architecture distinguishes tinnitus from hearing loss", the present format hides the "distinguishing" part in a way that could be improved.

Also, please clarify what genetic data was used for the "Genetic overlap analyses". Was this using the 39 top loci or did this use the entire GWAS data and what was the phenotype used? Tinnitus only or tinnitus + HD.

Phenotype: when cases were selected for tinnitus, were individuals with hearing difficulties excluded? So, if an individual had HD+ tinnitus, was she/he included (I assume, yes). Please clarify. How about cases that only scored for tinnitus (was their age distribution similar), what was the number of such cases?

Minor comment:

Introduction: This reviewer would recommend deleting references 20 and 21 aiming to identify coding variants in tinnitus. These studies are severely underpowered and do not meet high standards of genetic variant discovery.

Reviewer #3 (Remarks to the Author):

The manuscript titled "Genetic architecture distinguishes tinnitus from hearing loss" presents the largest GWAS and meta-analysis of multiple population cohorts to date that identified 39 tinnitus loci, including 37 not previously described. The authors identified shared but also distinct genetic signatures from hearing difficulty, with evidence of higher polygenicity. Expression analysis in tissues highlight significant enrichment of brain tissues. The noted correlation to not only hearing loss but also psychiatric disorders was quantified, as well as a scoping analysis of drugs and pathways. Emphasis of support for patients from multiple medical specialties is justified through this work.

The work completed in this study is extensive and very well carried out. It will be a remarkable contribution to the field. The methodology was elaborate and concisely described. I cannot comment heavily on the thoroughness of the various computational and statistical frameworks applied.

I have a few questions below, as a few things were slightly unclear. Apologies if some of these are rather straightforward:

-Extensive clinical heterogeneity concerning the subtypes of tinnitus have been discussed at length in the literature. Can the authors separate laterality, duration (acute or chronic) or other risk factors in these sub-types? How do the authors frame the results with stratification of sub-type being an aspect that other researchers have noted as critical for genetics studies?

-The authors uncovered 62 genome-wide significant genes in the European American group, including 16 not identified previously. The link between the 62 genes mentioned here and the 39 loci (Table 1) in the abstract is a bit disconnected and can be clarified a bit better.

-How can the authors separate hearing difficulty and tinnitus for several genes that are associated with genetic forms of hearing impairment, like COL11A1, among others? Of note, COL11A1 also causes non-syndromic deafness (line 179), DFNA37. NID2, to my knowledge, has only been linked to hearing function via GWAS and not through classical genetics studies, so in my view, this is not a well-

known gene (lines 182-183). It could certainly be deep in the supplement, but I could not find mention of NID2 in references 45-46.

How do the candidate genes that were identified in the Swedish study (refs 20, 21) compare with the findings here?

Figure 1a: the red line looks rather brown in my copy

Figure 2d: For what reason did the authors use liver as baseline for this analysis? The rationale should be explained.

REVIEWER COMMENTS

Reviewer #1 (Remarks to the Author):

Review

Clifford et al. follow-up on their previous GWAS study to better understand the differences between hearing loss and tinnitus from a genetic point of view. The authors perform a meta-analysis of the former datasets, and include non-European ancestries, to identify 39 loci. With enrichment analyses, they show that tinnitus finds multiple genes located in the brain, unlike hearing loss. They use single cell RNA sequencing data of the mouse adult cochlea to identify the peripheral contributors to tinnitus.

I congratulate the authors for this massive effort! This manuscript addresses a major issue in the field, and provides a significant advance. From a genetic point of view, it is seldom seen that GWAS try to disentangle two co-occurring conditions that are so closely related. Thus, this study will also provide important insights to the genetic community at large. The paper is very well written (with the exception of the introduction, half of which are results), follows a logical analytical flow, the majority of the statistics are sound. The scRNAseq maybe deserves some additional validation in order to exclude the possibility of false positives. This would require the analysis of additional cellular datasets. Finally, the summary statistics from the Million Veterans will be made available after publication – while that of the UKBB is accessible to the biobank itself: this needs to be addressed as for the sake of progress in science, and the needs of the reviewers to verify the claims, the data needs to be available now. With hopes these aspects can be addressed, then the manuscript would reach the expected quality.

We thank Reviewer 1 for these thoughtful comments and have addressed them in detail below.

Major comments:

1. The scRNAseq analysis uses a single data set by Hoa et al. This analysis is conditioned on liver, which is a novel way to address and improve the analysis. However, this protocol remains to become a standard. The authors should include the following datasets on separate confirmatory analyses. The analyses should include the traditional LDSC and MAGNA analyses separately.

a. Jean et al. Proc Natl Acad Sci U S A. 2023 Jun 27;120(26):e2221744120. doi: 10.1073/pnas.2221744120. Epub

b. Milon et al. Cell Rep. 2021 Sep 28;36(13):109758. doi: 10.1016/j.celrep.2021.109758.

c. Sun et al. Protein Cell. 2023 Apr 13;14(3):180-201. doi: 10.1093/procel/pwac058.

d. Xu et al. Front Cell Neurosci. 2022 Aug 18;16:962106. doi: 10.3389/fncel.2022.962106. eCollecBon 2022.

e. Iyer et al. Elife 2022 Nov 29;11:e79712. doi: 10.7554/eLife.79712.

We thank the reviewer for these suggestions. We made the following two modifications: 1. We have substantially revised the analysis protocol, following a recent paper by Trpchevska et al (AJHG 2023 <https://doi.org/10.1016/j.ajhg.2022.04.010>). Specifically, we are now conditioning on the average of the specific scRNAseq mouse cochlear cells (as opposed to the liver we have used before). As requested, we have also added a LDSC analysis to complement the MAGMA analysis. 2. We expanded the analyses to include a second, larger data set from Jean et al. (2023) and repeated the analyses described above. Our decision to give preference to the Jean et al. data over the other studies proposed by the reviewer is based on a preference for older mice (P20) with fully developed cochlea, and the accessibility and availability of pre-processed data. The corresponding Results and Method sections have been revised accordingly (updated Figure 2d, new Figure 2e and revised Supplementary Table 11).

2. Line 318 and subsequent paragraph: authors refer to the synaptopathy from Liberman and the article from Schaette. The manuscript from Liberman performed in mice has been cited nearly 1000 times but

never effectively reproduced. Likewise, the paper from Schaette has not yet been properly validated. I would minimize all references to synaptopathy and hidden hearing loss.

Thank you for this suggestion, as human studies on this topic have been problematic. The citations from Schaette and Liberman were removed, and “primary” in line 319 was changed to “suggested”. The references were all specified as mouse models. “Synaptopathy” and “hidden hearing loss” are not mentioned. The paragraph was changed to:

"Gene-based analysis categorizes multiple genes expressed within the synaptic area. Although this finding affects all synapses, it is consistent with findings of deafferentation of the inner hair cell synapse as a suggested area of cochlear injury.⁶⁷ This loss of synapses, occurring in aging mice as well as noise levels that do not induce objective audiogram changes, can be associated with a reduced auditory brainstem response Wave I, an indication of diminished connections from the cochlear spiral ganglion to the dorsal cochlear nucleus (DCN),⁶⁸ however human studies have been problematic.⁶⁹ Specifically, while this cochlear injury has been suggested as an initial injury, it has yet to be linked to the onset of tinnitus."

3. Authors should make the UKBB and MVP summary stats available now with a doi number accessible for the reviewers and released upon publication (e.g. zenodo). The GWAS data from both the UKBB and the MVP published in JAMA Otolaryngology Head and Neck Surgery was not accessible anywhere. This is quite a shame. The codes of the analysis should also be made available. I understand this is a major effort, but the auditory neuroscience community, which is not specialist in the field of genetics, would deserve having such methodologies available to advance the science on other fronts.

We apologize for the lack of releasing our version of the UKB tinnitus GWAS summary results from the JAMA Otolaryngology Head and Neck Surgery paper. These data have been available from the authors upon request, but as suggested by the reviewer, we have now released them on figshare (DOI: 10.6084/m9.figshare.24121281). The MVP GWAS summary data release has to follow the official MVP data access policy (<https://www.mvp.va.gov/pwa/discover-mvp-data>) and will be available on dbGAP (https://www.ncbi.nlm.nih.gov/projects/gap/cgi-bin/study.cgi?study_id=phs001672) after the embargo period has passed. As suggested, we now make the codes for the analysis available on Github (<https://github.com/nievergeltlab/Tinnitus>). These changes have been made to the Data availability paragraph.

Minor comments:

- The abstract does not convey the major findings from the paper, and ends blunt on a broad analytical statement. More of the major results should be reported here (what brain regions was tinnitus enriched... etc ..), with conclusions and perspectives on the knowledge the work has generated.

We have revised the abstract and added more specific results and conclusions.

Specifically, "... 39 tinnitus loci, including genes related to neuronal synapses and cochlear structural support. Applying novel analytic tools, we confirm a large number of shared variants, but also a distinct genetic architecture of tinnitus, with higher polygenicity and large proportion of variants not shared with hearing difficulty. Tissue expression analysis for tinnitus infers broad enrichment across most brain tissues, in contrast to hearing difficulty. Finally, tinnitus is not only correlated with hearing loss, but also with a spectrum of psychiatric disorders, providing potential new avenues for treatment."

- Half of the introduction, starting from the third paragraph, summarizes the results. Authors should keep the introduction to present the background knowledge in the field rather than stating what they found. Delete the whole paragraph (can be moved to the end of the discussion/conclusions) and include additional information on tinnitus and hearing loss, both environmental and genetic influencers.

We agree with the reviewers and have revised the introduction, added additional information on tinnitus and hearing loss in the new paragraphs 3 and 4, and removed most of the last paragraph that was summarizing the results.

“The high epidemiologic association of the two disorders points to an initial cochlear source of injury, whether the etiology is aging, noise, trauma, or otherwise, followed by cochlear deafferentation,²⁵ possibly through the loss of inner hair cell connections with spiral ganglion cells as an early event.²⁶ In the case of hearing loss, this deafferentation may drive neuroplasticity within the auditory pathway tonotopically to the auditory cortex, and is seen both on the audiogram as frequency specific and on diffusion tensor imaging (DTI) most commonly in the auditory cortex and inferior colliculus.²⁷

On the other hand, tinnitus may lead to different changes which will require coordination with genomic findings. Increased delta band activity is seen in the thalamus and auditory cortex.²⁸ The inferior colliculus shows changes ranging from hyperactivity secondary to decreased GABAergic inhibition, to altered levels of glutamic acid decarboxylase.²⁹ Imaging studies reveal changes in resting state and sound-evoked BOLD fMRI responses in hippocampus, insula, and prefrontal cortex, among others, indicative of changes in connectivity both within and outside the auditory pathway.³⁰ These neuroplastic changes occurring in the presence of tinnitus may indicate genetic variation at play in parts of the brain associated with perceptions, emotions, and cognition.”

- The first paragraph of the intro should mention the two studies that have shown that tinnitus is associated with increased latencies of the ABR Wave V (inferior colliculus). These studies are important because they are mutually confirmatory, despite the different analytical approaches (one stratifies by hyperacusis, the other one includes it as a variable in a statistical model)
 - o Hofmeier et al. Clin Transl Med. 2021 May;11(5):e378. doi: 10.1002/ctm2.378.
 - o Edvall et al. J Clin Invest. 2022 Mar 1;132(5):e155094. doi: 10.1172/JCI155094.Done; we have added: "Wave V of the auditory brain response (ABR) has demonstrated increased latency in constant tinnitus, indicative of delayed connectivity to the inferior colliculus within the auditory pathway.^{15,16} " to the first paragraph.

- Some of the references are wrong:

- o Line 51: Ref 13 is on Alzheimer’s risk, not on loss of inhibition

Line 51 references 3 (not 13), which is a general reference on tinnitus. We moved Reference 13 and explained it where we discuss neurologists’ role in tinnitus. Thus, “Neurologists treat cognitive disorders, which are associated with tinnitus, and ...

- o Line 61: Ref 16 is an adoption study, not a twin study

Added “Twin and adoption studies ...”

- o Line 59: The first study from Amanat was predominantly Spanish and replicated with a smaller set of a Swedish population.

Thank for this. We removed this sentence on request of Reviewer 2.

- Supplementary table 1 should present the numbers for the meta-analysis

We have added the numbers for the MVP meta-analysis to ST1 (rows 27-34).

- The very high prevalence of tinnitus in the MVP likely accounts for the high power of this analysis. Somewhere, the reader should understand that this is a very untypical cohort to work with.

In Discussion, second paragraph, we added: "Cohorts with a high prevalence of tinnitus, such as the MVP, may be particularly informative for gene discovery."

- Typo on Sup Table 16 (Adjutyed)
fixed

- Line 348: magnetoencephalography is a method – not a brain change.

This information has been changed to actual findings and a reference has been added. “Broad changes in gray and white matter, magnetoencephalography indication of tinnitus-frequency specific activity, and a difference in the default mode network (DMN) of tinnitus subjects have been identified, while hearing loss appears to be related more specifically to the cochlea and auditory pathway.^{65”}

In addition, the following references have been added:

1. Vasiliki Salvari, Daniela Korth, Evangelos Paraskevopoulos, Andreas Wollbrink, Daniela Ivansic, Orlando Guntinas-Lichius, Carsten Klingner, Christo Pantev, Christian Dobel, Tinnitus-frequency specific activity and connectivity: A MEG study. *NeuroImage: Clinical*, Volume 38, 2023, 103379, ISSN 2213-1582, <https://doi.org/10.1016/j.nicl.2023.103379>. (<https://www.sciencedirect.com/science/article/pii/S2213158223000682>)
2. Schmidt SA, Akrofi K, Carpenter-Thompson JR, Husain FT. Default mode, dorsal attention and auditory resting state networks exhibit differential functional connectivity in tinnitus and hearing loss. *PLoS One*. 2013 Oct 2;8(10):e76488. doi: 10.1371/journal.pone.0076488. PMID: 24098513; PMCID: PMC3788711.
3. Khan RA, Sutton BP, Tai Y, Schmidt SA, Shahsavarani S, Husain FT. A large-scale diffusion imaging study of tinnitus and hearing loss. *Sci Rep*. 2021;11(1):23395. Published 2021 Dec 3. doi:10.1038/s41598-021-02908-6

- Paragraph starting line 352: Please refer only to causal inferences (longitudinal or case-control), not cross-sectional. Discuss both the epidemiology of hearing loss and tinnitus.

We removed the following cross-sectional references:

Nondahl, D. M. et al. Tinnitus and its risk factors in the Beaver Dam offspring study.

Shargorodsky, J., Curhan, G. C. & Farwell, W. R. Prevalence and characteristics of tinnitus among US adults. Crönlein, T. et al. Insomnia in patients with chronic tinnitus: Cognitive and emotional distress as moderator variables.

However, since this paragraph refers specifically to the genetic overlap of tinnitus and other traits, and the analyses did not include the genetic overlap of hearing loss and other traits, we are not extending the discussion here to the epidemiology of hearing loss.

Reviewer #2 (Remarks to the Author):

The manuscript by Clifford et al describes a GWAS tinnitus performed in the VA and UKBB study sets. The phenotyping is based on self-reported questionnaires and/or ICD codes. The study samples are large and provide a very nice opportunity to tackle the genetic background of tinnitus. As many other complex diseases, tinnitus is not an independent disease but a symptom that associates with many other diseases/symptoms including hearing loss, depression and musculoskeletal traits. Thus, the dissection of “tinnitus specific gene loci” is challenging.

The manuscript includes comprehensive analyses. It also is, for the most part, clearly written.

However, some parts of the presentation in the main manuscript could be improved. Specifically, this reviewer would like to have a clearer presentation of loci that are “tinnitus specific/enriched” and their pleiotropy. Now most of the downstream analyses include hearing loss variants, which is logical due to the larger number of associated variants compared to “tinnitus specific loci”. However, this potentially also misses some interesting aspects.

We thank Reviewer 2 for these thoughtful comments and have addressed them in detail below.

- Figure 3 presents the analytical attempt to distinguish between tinnitus and HD loci. This reviewer finds this part of the paper the most interesting one. Yet, reading the results is not easy. While Manhattan plots are visually easy to grasp, they are not very informative in this setting. **A table would be a better format.** Now the actual results and the table is in the Supplement. This reviewer would recommend a simplified table produced from Supplementary tables 16 and 17. The table could also include the nearest gene/genes around the lead variant and pleiotropy analyses of the lead variant.

We thank the reviewer for these comments. We have attempted to make the reading of the results easier by combining Supplementary Tables 16 and 17 into one comprehensive table. This substantially revised Table 16 also includes genes around the lead variants, and now highlights disorder-specific loci. However, due to the large size of this Table, we have included it in the Supplement and kept Figure 3e, which is a graphical representation of simplified results.

- Also, a pleiotropy table of the main Table 1 as a Supplementary table might be helpful. The concern of this reviewer is that the current chapter “Genetic overlap of tinnitus and other traits” has a potential bias towards mental health due to the resources used for the analysis (especially the PGC dataset). This might miss some of the potential genetic correlations with autoimmune diseases, skeletal disorders and hypothyroidism. Was a locus specific PheWAS analysis performed using VA and UKBB datasets?

We now include a pleiotropy analysis in the supplementary Table 17, which includes a comprehensive collection of traits across many domains. The GWAS catalogue used includes the UKB dataset, but not the MVP.

In summary from the above, the current form of the presentations does not quite address the title “Genetic architecture distinguishes tinnitus from hearing loss”, the present format hides the “distinguishing” part in a way that could be improved.

Whereas the first part of the paper is focused on identification of risk loci and differences between tinnitus and HD, the main differences between the two disorders are found in the genetic architecture, which refers to the second part to the paper, namely the differences in polygenicity, discoverability, and gene expression in the brain. The title is summarizing our main findings. We did, however, improve the presentation of the distinguishing risk loci (see above).

- Also, please clarify what genetic data was used for the “Genetic overlap analyses”. Was this using the 39 top loci or did this use the entire GWAS data and what was the phenotype used? Tinnitus only or tinnitus + HD.

These analyses used the entire GWAS data from the main tinnitus GWAS in European ancestry. We added this information to the result section.

Phenotype: when cases were selected for tinnitus, were individuals with hearing difficulties excluded? So, if an individual had HD+ tinnitus, was she/he included (I assume, yes). Please clarify. How about cases that only scored for tinnitus (was their age distribution similar), what was the number of such cases?

Thank you for these questions, we agree that adding this information will improve the manuscript. To answer these questions, we have now stratified the subjects by 'hearing difficulty status' in the revised Supplementary Table 1. Indeed, we did not exclude subjects with HD from the tinnitus cases or controls, and remained agnostic to hearing difficulty status for the main tinnitus analyses.

As expected, tinnitus subjects with hearing difficulty were significantly older than subjects without hearing difficulty. We have added this information in footnote d of the revised Supplementary Table 1.

"UKB: Age for tinnitus cases with hearing difficulty (mean = 60.34, SD = 7.54) was significantly higher than age for tinnitus cases without hearing difficulty (mean = 57.69, SD = 8.23; $p < 0.0000$)

MVP: Age for tinnitus cases with hearing difficulty (mean = 64.84, SD = 11.73) was significantly higher than age for tinnitus cases without hearing difficulty (mean = 55.07, SD =14.88; $p < 0.00001$)".

Minor comment:

Introduction: This reviewer would recommend deleting references 20 and 21 aiming to identify coding variants in tinnitus. These studies are severely underpowered and do not meet high standards of genetic variant discovery.

Thank you. Removed the following sentence: "while sequencing studies investigating rare variant burdens in predominantly Spanish patients with severe tinnitus identified a small number of candidate genes.^{20,21}"

Reviewer #3 (Remarks to the Author):

The manuscript titled "Genetic architecture distinguishes tinnitus from hearing loss" presents the largest GWAS and meta-analysis of multiple population cohorts to date that identified 39 tinnitus loci, including 37 not previously described. The authors identified shared but also distinct genetic signatures from hearing difficulty, with evidence of higher polygenicity. Expression analysis in tissues highlight significant enrichment of brain tissues. The noted correlation to not only hearing loss but also psychiatric disorders was quantified, as well as a scoping analysis of drugs and pathways. Emphasis of support for patients from multiple medical specialties is justified through this work.

The work completed in this study is extensive and very well carried out. It will be a remarkable contribution to the field. The methodology was elaborate and concisely described. I cannot comment heavily on the thoroughness of the various computational and statistical frameworks applied. I have a few questions below, as a few things were slightly unclear. Apologies if some of these are rather straightforward.

We thank Reviewer 3 for these thoughtful comments and have addressed them in detail below.

-Extensive clinical heterogeneity concerning the subtypes of tinnitus have been discussed at length in the literature. Can the authors separate laterality, duration (acute or chronic) or other risk factors in these sub-types? How do the authors frame the results with stratification of sub-type being an aspect that other researchers have noted as critical for genetics studies?

We agree with the reviewer that this information would be very useful. Unfortunately, ICD's and questionnaires did not separate acute versus chronic, nor subtypes. We have noted in the last paragraph that "Further work will attempt to characterize tinnitus with various aspects of audiology available in the medical record." and are currently working on this with the hope to be able to address tinnitus sub-types to some degree.

-The authors uncovered 62 genome-wide significant genes in the European American group, including 16 not identified previously. The link between the 62 genes mentioned here and the 39 loci (Table 1) in the abstract is a bit disconnected and can be clarified a bit better.

We agree and changed the text to: "As an alternative strategy to SNP-based analyses, gene-based analyses capture all of the potential risk-conferring variations within a gene and allow integration with gene expression data." ..."Gene-based analyses identified 62 GWS genes, including 16 genes not identified by mapping approaches applied to the EA or cross-ancestry GWASs".

-How can the authors separate hearing difficulty and tinnitus for several genes that are associated with genetic forms of hearing impairment, like COL11A1, among others? Of note, COL11A1 also causes non-syndromic deafness (line 179), DFNA37.

The sentence was changed to "... a known non-syndromic and syndromic deafness gene ..."

NID2, to my knowledge, has only been linked to hearing function via GWAS and not through classical genetics studies, so in my view, this is not a well-known gene (lines 182-183). It could certainly be deep in the supplement, but I could not find mention of NID2 in references 45-46.

Thank you for this. This line was changed to:

“NID2, identified in previous hearing GWAS’, binds to collagens I and IV in the basement membrane.”

How do the candidate genes that were identified in the Swedish study (refs 20, 21) compare with the findings here?

These references were removed, as they were underpowered studies.

Figure 1a: the red line looks rather brown in my copy

We renamed the red line brown line.

Figure 2d: For what reason did the authors use liver as baseline for this analysis? The rationale should be explained.

As requested by Reviewer 1, we substantially revised this analysis and the revised version is not using the liver as a baseline (see also comments to Rev. 1 for more details).

REVIEWERS' COMMENTS

Reviewer #1 (Remarks to the Author):

The authors have addressed very largely the comments I made, and I am pleased to see the current version!

Some minor points:

- I could not access the figshare sumstats from the UKBB - the link provided does not exist.
- The Github codes possess the code related to the UKBB data, but not the MVP - please clarify. The readme file is empty and needs proper guidance through the released codes, and upon publication a link to the paper.

As shown by Hoffmeier et al. and the paper by Edvall et al., the results on tinnitus are highly confounded with hyperacusis (see Cederroth et al., JCM, 2020 - <https://www.mdpi.com/2077-0383/9/8/2412>). The discussion should address (I know the tight limitations in space) at least the fact that future studies will need to disentangle the genetics underlying the two (should such information be available in the two cohorts?).

Reviewer #2 (Remarks to the Author):

The authors have addressed this reviewer's concerns.

Response to Reviewers

Reviewer #1 (Remarks to the Author):

The authors have addressed very largely the comments I made, and I am pleased to see the current version!

We thank Reviewer 1 for their thoughtful comments for both revisions and have addressed the remaining items below.

Some minor points:

- I could not access the figshare sumstats from the UKBB - the link provided does not exist.
- The Github codes possess the code related to the UKBB data, but not the MVP - please clarify. The readme file is empty and needs proper guidance through the released codes, and upon publication a link to the paper.

We have updated the data availability statement accordingly and provide the link here (<https://doi.org/10.6084/m9.figshare.24121281>). The Github repository now has the code for both the UKB and MVP datasets.

As shown by Hoffmeier et al. and the paper by Edvall et al., the results on tinnitus are highly confounded with hyperacusis (see Cederroth et al., JCM, 2020 - <https://www.mdpi.com/2077-0383/9/8/2412>). The discussion should address (I know the tight limitations in space) at least the fact that future studies will need to disentangle the genetics underlying the two (should such information be available in the two cohorts?).

We have addressed this comment in our introduction. However, the lack of this diagnosis in our MVP database (450 out of 308,879 European ancestry participants) does not allow us to further disentangle the genetics between the two highly correlated disorders.

Reviewer #2 (Remarks to the Author):

The authors have addressed this reviewer's concerns.

We thank Reviewer 2 for their thoughtful comments on the revision.